



# Temporal patterns and biophysical controls on methane emissions from a small eutrophic reservoir: insights from two years of eddy covariance monitoring

Sarah Waldo[1*], Jake J. Beaulieu[1], William Barnett[2], David A. Balz[3], Michael J. Vanni[4], Tanner Williamson[4], and John T. Walker[5]

[1]Center for Environmental Measurements and Modeling, United States Environmental Protection Agency, Office of Research and Development, Cincinnati, 45268, USA
[2]Neptune and Company, Inc., Lakewood, 80215, USA
[3]Pegasus Technical Services, Cincinnati, 45268, USA
[4]Miami University, Department of Biology, Oxford, 45056, USA
[5]Center for Environmental Measurements and Modeling, United States Environmental Protection Agency, Office of Research and Development, Durham, 27709, USA

*Correspondence to*: Sarah Waldo (sarahrwaldo@gmail.com)
*Currently at United States Environmental Protection Agency, Region 10, Seattle, 98101, USA*

**Abstract.** Waters impounded behind dams (i.e. reservoirs) are important sources of greenhouses gases, especially methane ($CH_4$), but their contribution is not well constrained due to high spatial and temporal variability, limitations in monitoring methods to characterize hot spot and hot moment emissions, and the limited number of studies that investigate diurnal, seasonal, and interannual patterns in emissions. In this study, we investigate the temporal patterns and biophysical drivers of $CH_4$ emissions from Acton Lake, a small eutrophic reservoir, using a combination of methods: eddy covariance monitoring, continuous warm-season ebullition measurements, spatial emission surveys, and measurements of key drivers of $CH_4$ production and emission. We used an artificial neural network to gap-fill the eddy covariance time series and to explore the relative importance of biophysical drivers on the inter-annual timescale. Acton Lake had cumulative areal emission rates of $40.6 \pm 5.9$ and $71.4 \pm 4.2$ g $CH_4$ m$^{-2}$ in 2017 and 2018, respectively, or $97.4 \pm 14$ and $171 \pm 10$ Mg $CH_4$ in 2017 and 2018 across the whole 2.4 km$^2$ area of the lake. The main difference between years was a period of elevated emissions lasting less than two weeks in the spring of 2018, which contributed 17% of the total annual emissions, and was likely due to favourable sediment temperature and algal carbon substrate availability in 2018 compared to 2017. $CH_4$ emissions only displayed diurnal patterns 18.5% of the monitoring period, suggesting that factors that do not follow a diurnal pattern (e.g. substrate availability) may be driving emissions. Combining spatially extensive measurements with temporally continuous monitoring enabled us to quantify aspects of the spatial and temporal variability in $CH_4$ emission. We found that the relationships between $CH_4$ emissions and sediment T depended on location within the reservoir and observed a clear spatio-temporal offset in maximum $CH_4$ emissions as a function of reservoir depth. These findings suggest a strong spatial pattern in $CH_4$ biogeochemistry within this relatively small (2.4 km$^2$) reservoir. In addressing the need for a better understanding of GHG emissions from reservoirs, there is a trade-off in intensive measurement of one water body versus short-term and/or spatially limited measurements in many water bodies. The insights from multi-year, continuous, spatially extensive studies like this one can be used to inform both the study design and emission upscaling from spatially or temporally limited results, specifically the importance of trophic status and intra-lake variability in assumptions about upscaling $CH_4$ emissions.



## 1 Introduction

Reservoirs are a globally important source of methane ($CH_4$) and other greenhouse gases (GHG) to the atmosphere, with recent estimates attributing 773 Tg carbon dioxide equivalents ($CO_2$-e) per year to reservoir surface emissions, nearly 80% as $CH_4$

(Deemer et al., 2016). The dominance of $CH_4$ in reservoir GHG budgets is due to the combination of gross $CH_4$ emissions and methane's large warming potential relative to $CO_2$. Reservoir $CH_4$ emissions have been estimated to be equivalent to roughly half of the global $CH_4$ burden from rice cultivation (~1200 Tg $CO_2$-e $yr^{-1}$, Ciais et al., 2013). Inland waters (lakes, rivers, and reservoirs) can be hot spots for microbial decomposition of organic matter, and respiration from these waters globally may offset the terrestrial carbon sink by up to 60% (Cole et al., 2007; Ciais et al., 2013). The carbon dynamics of reservoirs are of

special interest for several reasons. Reservoirs generally receive more sediment input (hence organic C) from their watershed than comparable lakes, as they tend to be located lower in the landscape and have a larger ratio of catchment area to surface area (Hayes et al., 2017). Reservoirs also tend to drain watersheds with more agricultural or urban land use than the natural lake watersheds (Thornton et al., 1990). The distribution of lakes and reservoirs across the United States is such that in many parts of the country total lentic surface area is dominated by reservoirs. Furthermore, emissions from reservoirs are considered

anthropogenic and thus should be included in national greenhouse gas (GHG) emission inventories reported to the United Nations (Lovelock et al., 2019).

Emissions of GHGs from reservoirs are highly variable in space in time, making the reservoir GHG budgets difficult to constrain. This is especially true for $CH_4$, for which ebullition, or bubbling, is an important emission pathway. $CH_4$ is produced in reservoir sediments by methanogens. Some of this $CH_4$ dissolves into the water column where it may be oxidized into carbon

dioxide by methanotrophs or may diffuse to the atmosphere. Methane may also accumulate as bubbles in the sediment until the buoyant force of the gas bubble overcomes the overlying static pressure. The rate of $CH_4$ ebullition is affected by several biological and physical factors including carbon substrate availability, sediment temperature, oxygen availability, turbulence and overlying pressure (Tuser et al., 2017). Thus, ebullition is highly variable in space and time (Wik et al., 2016).

Although the body of knowledge on $CH_4$ emissions from inland waters has grown considerably over the past decades, the high

degree of spatial and temporal variability in emissions, coupled with limitations in monitoring methods, mean that many questions about reservoir emission behaviour remain. Recent studies have highlighted the importance of interannual patterns (Room et al., 2014), seasonal patterns (Yvon-Durocher et al., 2014), diurnal patterns (Podgrajsek et al., 2014, Deshmukh et al., 2014), lake-zone spatial patterns (Juutienen et al., 2009; DelSontro et al., 2011; Maeck et al., 2013), and the relative contributions of hot-spots (Wik et al., 2016, Beaulieu et al. 2016) and hot-moments (Bastien et al., 2011, Demarty et al., 2011,

Jammet et al., 2015, Beaulieu et al., 2018, Harrison et al., 2018) in accurately characterizing lake and reservoir $CH_4$ emissions. Use of micrometeorological methods such as eddy covariance (EC) to monitor reservoir $CH_4$ fluxes ($F_{CH4}$) can address many of the monitoring challenges by providing pseudo-continuous, long-term, spatially integrated flux measurements. As described in the synthesis by Deemer et al. (2016), studies using the spatially integrated methods of EC or acoustic monitoring reported higher $F_{CH4}$ than the median $F_{CH4}$ from studies using methods with less spatial and temporal coverage (e.g. floating chambers,

thin boundary layer, inverted funnels), consistent with the finding that low sampling coverage in stochastic systems leads to underestimation (Wik et al., 2016). The recent FLUXNET-$CH_4$ synthesis (Knox et al., 2019) of long-term (>1 year) EC



monitoring of $F_{CH4}$ had only two open water sites among the 60 included. A low-power open-path $CH_4$ sensor capable of making measurements for EC has only been available since circa 2011 (McDermitt et al., 2011), and using micrometeorological techniques to measure fluxes over open water (vs. land) can be difficult due to siting, footprint, and boundary layer turbulence considerations (Kenny et al., 2017, Higgins et al., 2013, Sahlee et al., 2014). Thus, relatively few studies have used EC to

characterize $F_{CH4}$ over inland waters (Jammet et al., 2015, Jammet et al. 2017, Deshmukh et al., 2014, Eugster et al., 2011, Schubert at al., 2012, Podgrajsek et al., 2014a, Podgrasek et al., 2014b, Beaulieu et al., 2018). To our knowledge, this study is only the second to report pseudo-continuous, multi-year $F_{CH4}$ results over open-water, and the first to report long-term $F_{CH4}$ over open-water in a temperate region, for a eutrophic system, and for a reservoir.

In this study, we investigate biophysical drivers of $CH_4$ emissions from a midlatitude reservoir at multiple temporal and spatial

scales by combining two years of continuous $F_{CH4}$ measured over the shallower inlet region of the reservoir via EC, warm-season continuous ebullition measurements at a shallow and deep site, spatially balanced $CH_4$ emission surveys, and measurements of key drivers of $CH_4$ production and emission. We expected to see elevated emissions in the shallower inlet area relative to the rest of the reservoir based on previous work (Beaulieu et al., 2016), and designed our investigation to leverage both spatial and temporal coverage in characterizing reservoir $F_{CH4}$. We use an artificial neural network to gap-fill the

EC timeseries and to provide insight into the most important biophysical drivers of $F_{CH4}$, notably sediment temperature, overlying static pressure, and water column mixing. Insights into temporal patterns and biophysical drivers of $F_{CH4}$ from individual water bodies will provide important constraints on the contribution of inland waters to the global GHG budget.

## 2. Methods

### 2.1 Site description

Acton Lake is a small hypereutrophic reservoir located in southwestern Ohio (39.57 N, 84.74 W, 262 masl, Fig. 1a). The dam was constructed in 1956 and the lake and surrounding state park have been managed by the Ohio Department of Natural Resources since 1957. The lake's surface area is 2.4 km$^2$, it has a maximum depth of ~8 m, and the area near the dam undergoes thermal stratification in the summer. Although Acton Lake is immediately surrounded by a forested state park, land use in its watershed is >80% agricultural, with the majority used for intensive row cropping (Renwick et al., 2018). We used four main

methods to monitor $CH_4$ fluxes ($F_{CH4}$) from Acton Lake during the period between 1 February 2017 thru 1 December 2018: (1) the EC technique, (2) continuous ebullition monitoring with active funnel traps, (3) bi-weekly chamber measurements of diffusive emissions, and (4) spatially extensive surveys. The locations of the EC tower sites, active funnel trap/bi-weekly chamber measurement sites, and spatially extensive survey sites are depicted in Fig. 1a; the cumulative footprint probability distribution of the two flux tower sites is shown in Fig. 1b. The EC instrumentation was sited in the shallow region of Acton

Lake due to logistical constraints related to both tower installation and boat traffic in the reservoir. How the methods were used in this study is summarized in Table 1. We used auxiliary meteorological and limnological measurements from stream gauging stations, a weather station, and thermistor string maintained by the Miami University (Renwick et al. 2018; Andersen et al. 2020), the locations of which are also shown in Fig. 1a.



## 2.2 Eddy covariance flux measurements

This site is registered as AmeriFlux site US-Act; information about the site as well as the flux data presented in this study are available online (https://ameriflux.lbl.gov/sites/siteinfo/US-Act). The EC instrumentation consisted of an ultrasonic anemometer to measure 3-D wind speed and direction (Model 81000, R.M. Young Company, Traverse City, MI, USA) and

open path infrared gas analyzers (IRGAs) for measuring the number density of $CH_4$ (LI-7700), and $CO_2$ and water vapor (LI-7500A, LiCor Biosciences, Lincoln, NE, USA). The EC data streams were recorded at 10-Hz by a data logger (LI-7550, LiCor Biosciences, Lincoln, NE, USA), which was also equipped with a temperature sensor and a pressure transducer. The EC system was deployed from a dock piling 20 m from the northwestern shore of Acton Lake from 1 February 2017 thru 14 April 2018 ("EC S-1" in Fig. 1). The instruments were brought to the lab for calibration and maintenance on 15 April 2018, then redeployed

on a tower installed into the lake sediment in the northeast corner of the lake on 5 May 2018 ("EC S-2" in Fig. 1). Images of the EC system at each deployment location are included in the SI (Fig. S1). In addition to the EC setup, the flux tower was equipped with a net radiometer (NRLite2, Kipp and Zonen, Delft, The Netherlands), a cellular modem for remote communication (AirLink, Campbell Scientific, Logan, UT, USA), and a time-lapse camera (WCT-00125 TimelapseCam, WingScapes, Calera, AL, USA). The system was powered by solar panels and a battery bank regulated via a solar charge

controller (SunSaver, Morningstar Corporation, Newtown, PA, USA). All components of the EC system were run on a 12V system until relocation to the aquatic tower, when the EC setup (LI-7700, LI-7500A, RMYoung, and LI-7500) was retrofitted to run on 24V.

The raw 10-Hz EC data was processed into 30-minute fluxes using the software EddyPro v. 6.2 (Licor Biosciences, Lincoln, NE, USA). We used measurements of water depth from the Miami University weather station to determine instrument height

above water surface on an hourly timestep, integrated into the flux processing as a dynamic metadata file. Additional processing steps followed community standards and included filtering the 10 Hz $CO_2$ measurements when $CO_2$ signal strength was <70, double coordinate rotation, block averaging, time lag compensation using covariance maximization, WPL density correction (Webb et al., 1980), and correction for high-pass and low-pass filtering effects (Moncrieff et al., 2004; Moncrieff et al., 1997). The area contributing to the measured flux was characterized for both sites using the online two-dimensional flux-footprint

prediction tool (Kljun et al., 2015). We used R for postprocessing, and the code is available on GitHub (https://github.com/USEPA/actonEC). The 30-minute fluxes were rejected when the period did not pass the tests for stationarity and developed turbulent conditions (QC level 2 per the integrated scale of Foken et al., 2004). EC S-1 fluxes were further filtered for periods when winds were from the shore (between 195° and 330°); at EC S-2 we filtered for periods of low turbulence using a friction velocity ($u_{star}$) threshold of 0.07 m s$^{-1}$, based on the site-specific relationship between $u_{star}$ and fluxes

of $CH_4$ and $CO_2$ (Aubinet et al., 2012). We define "acceptable" data or "acceptance rate" as those data meeting the EC QA/QC requirements, while "data coverage" includes non-operability due to power or instrument failures.

## 2.3 Active funnel trap ebullition measurements

The active funnel traps (AFT) were based on the design of Varadharajan et al. (2010) and have been previously described by Beaulieu et al. (2018). Briefly, they consisted of a 0.3 m$^2$ funnel attached to a rigid tubing gas collection chamber equipped

with a differential pressure sensor to monitor accumulated gas volume on a 5-minute timestep. We modified the Varadharajan design by incorporating siphons that auto-purge the collected bubble gas and refill the tubing volume with water. This



modification keeps the AFTs from becoming filled with gas, allowing them to make useful measurements for longer periods of time. Trap gas samples were collected bi-weekly and analysed via a gas chromatograph equipped with a flame ionization detector (Bruker 450 GC, USA) to determine the composition of the bubble gas. The active trap data reduction followed the method described in Varadharajan et al. (2010) and Varadharajan and Hemond (2012). Circuit calibration to determine the

relationship between voltage and height was performed pre- and post-trap deployment in the 2017 field season, and post-deployment in the 2018 field season. The volume of gas in the trap is calculated as:

$$AFT_{vol} = (Circ_{volt}*m + b)*\pi \frac{AFT_d^2}{2} \quad (1)$$

where $AFT_{vol}$ is the volume of gas in the funnel trap, $Circ_{volt}$ is the voltage output from the differential pressure sensor, $m$ and $b$ are the sensor-specific laboratory calibration multiplier and offset coefficients, and $AFT_d$ is the diameter of the funnel tubing.

Figure S2 shows a time series of the active trap volume measurements. We used a 12-point moving average (60 min) to smooth the gas volumes and minimize noise. Periods with known issues were filtered out of the dataset (e.g. power issues, trap drift from target location, etc.), as were large negative fluxes that reflected siphon purges. Following Varadharajan and Hemond (2012), we calculated fluxes on multiple time-bin widths (30-min, 1, 2, 6, 12, 24, 48 hr) but used the 2-hr rolling timestep for calculating the flux used in our final analysis:

$$F_{CH4_{eb}} = \frac{AFT_{vol}[CH_4]}{(T_f - T_i)A_F} \quad (2)$$

where $AFT_{vol}$ is the volume of gas in the trap (l), $[CH_4]$ is the $CH_4$ concentration in the bubble gas (mg $CH_4$ l$^{-1}$), $T_f - T_i$ is the elapsed time (s), and $A_F$ is the cross-sectional area of the funnel (m$^2$). The AFT data reduction was performed in R and the scripts are available online (https://github.com/USEPA/actonEC).

**2.4 Chamber diffusion measurements**

Diffusive $F_{CH4}$ was measured with a floating chamber biweekly at two sites during the field season. We used a rectangular, round-ended aluminum chamber with external polyvinyl chloride floats and a headspace fan, based on the CSIRO chamber described in Zhao et al. (2015). An ultra-portable greenhouse gas analyzer (UGGA, PN: 915-0011, ABB, Los Gatos, CA) monitored the change in $CH_4$ mixing ratio in the chamber headspace over the duration of the chamber deployment (> 1 - 5

min), measuring at 1Hz and recording an averaged measurement every 5 s. We monitored the real-time UGGA time series to prevent ebullitive emissions from overwhelming the diffusive emission measurements. If a spike in $CH_4$ concentration was detected, we re-set the chamber. The floating chamber data reduction method has been described in detail in Beaulieu et al. (2016). Briefly, we used the following equation to calculate diffusive fluxes (moles m$^{-2}$ s$^{-1}$):

$$F_{gas,D} = \frac{d\chi_{gas}}{dt}\left(\frac{V}{A}\right)\left(\frac{P}{RT}\right) \quad (3)$$

where $d\chi_{gas}/dt$ is the rate of change of the mixing ratio of $CH_4$ in the chamber headspace (ppm s$^{-1}$), $V$ is the chamber volume (m$^3$), $A$ is the chamber surface area (m$^2$), $P$ is the pressure in the chamber headspace, $R$ is the universal gas constant, and $T$ is the temperature in the chamber headspace. The rate of change $d\chi_{gas}/dt$ for each chamber deployment was determined via fitting linear and non-linear models to the dataset and using Akaike information criterion (AIC) to choose the more appropriate model. Only models with an $r^2 > 0.9$ were retained. Data analysis and reduction was performed using R, and the scripts are available

online (https://github.com/USEPA/actonEC).





### 2.5 Water measurements

Water temperature depth profiles were recorded continuously at two sites close to U-14 and U-12 (Fig. 1) using thermistors. At the shallow site (U-14) a string of seven thermistors (RBRsoloT, RBR Ltd., Ottowa, ON, Canada) were deployed at 0.1, 0.25, 0.5, 0.75, 1, 1.5 m below the air-water interface and at the sediment-water interface. We used this temperature profile to
characterize water column stability in the footprint of the EC flux measurements based on the Brunt-Vaisala buoyancy frequency using the R package rLakeAnalyzer (Winslow et al., 2019). The Brunt-Vaisala buoyancy frequency was used as to indicate water column stability. It represents the frequency at which a parcel of fluid will oscillate when displaced vertically, a measure of resistance to mixing.  A high oscillation frequency indicates strong resistance to mixing, whereas a low frequency indicates little resistance to mixing.  At the deep site (U-12), sondes measuring temperature (Pro ODO, YSI Incorporated,
Yellow Springs, OH, USA) were deployed at 0.1, 0.5, 1, 1.5, 2, 3, 4, 5, 6, 7, and 8 m below the air-water interface. Water temperature, specific conductivity, dissolved oxygen, pH, and chlorophyll a were measured biweekly with a YSI Multiparameter sonde at 0.1 and 1.5 m below surface at the shallow site (U-14), and 0.1, 1, 2, 3, 4, 5, 6, 7, and 8 m below surface at the deep site (U-12). Water samples for chlorophyll analysis were collected by Miami University near the reservoir inlet. Water samples were collected with an integrated tube sampler from the water surface to the euphotic zone depth.
Chlorophyll samples were collected on 1.0 μm glass fibre filters and frozen at -20°C in opaque containers until processed. They were extracted in 95% ethanol for 24 h and analysed with a TD-700 (Turner Designs, San Jose, CA, USA).

Dissolved gas surface and profile samples were collected biweekly from both U-12 and U-14 using the headspace equilibration method. We collected water samples at depths of 0.1, 2, 4, 6, and 7 m at U-12 and at 0.1, 0.75, and 1.3 m at U-14. Using a 140-ml plastic syringe with a 2-way stopcock, we added 25 ml of ultra-high-purity helium to a syringe, then added 115 ml of sample
water and agitated all samples for 5 minutes. We then transferred the headspace gas to pre-evacuated 12-ml glass vials topped with a silicone-coated Teflon septum stacked on top of a chlorobutyl septa (Labco Ltd., UK). The headspace gas samples were analysed using gas chromatography (see section 2.3) to determine the $CH_4$ composition, and the dissolved $CH_4$ concentrations were calculated using measured headspace composition and the temperature-specific Bunsen solubility coefficients (Yamamoto et al., 1976). Full documentation of the calculations is available at the National Ecological Observatory Network's GitHub
repository (https://github.com/NEONScience/NEON-dissolved-gas).

### 2.6  Whole-lake surveys

We conducted six surveys of Acton Lake over the summers of 2017 and 2018 to estimate whole-lake $F_{CH4}$. The fifteen sample collection sites (Fig. 1, light blue circles), were determined using a generalized random tessellation survey (GRTS) design
(Stevens and Olsen 2004; Olsen et al. 2012), a probability design that has been shown to reduce uncertainty relative to other designs (Beaulieu et al., 2016). At each site, we measured $CH_4$ diffusion, $CH_4$ ebullition, and surface water quality parameters. Survey measurements of diffusive $F_{CH4}$ were conducted with floating chambers in the same manner as described in section 2.4. Survey measurements of ebullitive $F_{CH4}$ were conducted with passive funnel traps (PFTs) deployed overnight (>15 hours). The PFTs are a simplified version of the AFTs described in section 2.3: they consist of a 0.3 $m^2$ funnel attached to a section of
tubing for gas collection, but do not have a pressure sensor or siphon. Upon retrieval, the total time of deployment and total volume of gas in the tubing were recorded, and three 25 mL samples of the gas were collected for gas composition analysis via



gas chromatograph (see section 2.3). Ebullitive $F_{CH4}$ from the PFTs was also calculated using Equation 2 (section 2.3), but the trap volume was determined by direct measurement of the collected gas, and $T_f – T_i$ is defined as the deployment period. Dissolved gas sample collection and depth profiles of water quality parameters were taken at one deep site (U-12) and one shallow site (U-14) during each whole-lake survey.

### 2.7 Gap-filling and up-scaling

Machine learning tools, including artificial neural networks (ANNs), are becoming more commonly used in studies investigating biogeochemical cycling due to their flexibility dealing with multiple non-linear drivers. We developed an ANN to gap-fill the 30-minute $F_{CH4}$ time series using predictor variables with biophysical links to $CH_4$ production and emission:

overlying static pressure, change in static pressure, sediment temperature (sedT), air temperature, latent heat flux (LE), sensible heat (H), wind speed, $u_{star}$ (friction velocity, a measure of turbulence), and photosynthetically active radiation. We also included indicators for the tower location, hour of day, and day of year as drivers. Gaps in the sedT, air temperature, wind speed, wind direction, and static pressure time series were filled using observations from a nearby weather station. Gaps in LE, H, and $u_{star}$ were gap filled using the mean diurnal course function from the R package REddyProc (Wutzler et al., 2019) on the 30-minute

timestep. We used k-means clustering to assign ten clusters before selecting the training, testing, and validation datasets. The cluster assignments allowed us to select subsets with probabilities proportional to the clusters, ensuring that the clusters were not over- or underrepresented as a result of the splits. We employed a selective ensemble approach to optimize the ANN model performance, using the R package nnet (Venables and Ripley, 2020). Each ANN ensemble included models with 5-20 layers and 50 different starting weights, for a total of 800 model results. The top 100 models were selected based on the testing $R^2$

results, then the median $CH_4$ value from the best 100 models was used as the predicted flux. To characterize both sampling and model uncertainty, we replicated this procedure with 20 resamplings of the data. For each half hourly $F_{CH4}$ we calculated the median predicted value of the best 100 models in each of the 20 ensembles of 800 models (c.f. Knox et al., 2016). Missing half hourly $F_{CH4}$ values were gap filled using the median of the medians from the 20 ensembles. ANN modelling and gap-filling was performed in R and the scripts are available online (Barnett et al., 2021).

For this analysis, we separated the year into different seasons, categorizing November thru March as "winter", or the cold season, and May thru September as "summer", or the warm season. We refer to April and October as the "shoulder" season. The spring burst period is defined as 24 May thru 4 June.

### 2.8  Uncertainty Analysis

We parameterized the uncertainty in the EC time series of $F_{CH4}$ using three different measures: the random measurement error, the bias error of the gap-filled dataset, and the 95% confidence intervals of the gap-filled dataset. The random measurement error is calculated from the variance of the covariance (Finkelstein and Sims, 2001), and reflects instrument noise, variation in footprint over a given 30-minute flux integration period, and the stochastic nature of turbulence. As described in Jammet et al.,

(2017), the random error decreases with increasing dataset size and is negligible at the resolution of cumulative annual fluxes but can be substantial for individual flux measurements (Richardson et al., 2006; Moncrieff et al., 1996). The random error was





calculated as part of the EddyPro processing, and we report the summary statistics in section 3.2. Unlike random errors, systematic biases can accumulate to affect the cumulative seasonal or annual flux. Although the measurement bias cannot be quantified, we calculated the systematic bias in the annual fluxes due to gap-filling following Moffat et al. (2007) and Jammet et al. (2017):

$$BE = \frac{1}{N} \sum (p_i - o_i) \quad (5)$$

where N is the number of values in the validation time series, p are the values predicted by the ANN, and o are the observed values in the validation time series. The bias error was multiplied by the total number of gap-filled values to obtain the total annual bias. We calculated the 95% confidence interval of the gap-filled dataset using the distribution of the 20 ANN medians extracted from the 20 resamplings, which consider both sample and model uncertainty (Knox et al., 2016).

We used root-sum-squared error propagation of the error in $AFT_{vol}$ and $[CH_4]$ to characterize the uncertainty in ebullitive $F_{CH4}$ measured by the AFTs. Compared to error in $AFT_{vol}$, the error contribution from other terms in Eqn 2 was negligible. As described in Varadharajan et al. (2010), we propagated the error in $m$, offset, and electronic noise through Eqn 1, adding a 2-ml dead volume error each time the AFTs flushed to account for gas that could be trapped in the fittings at the top of the collection chamber. Our mean slope and slope error were similar to those reported in the Varadharajan methods paper (31 and

0.31, respectively, compared to 28 and 0.5); the mean ($V_{zero}$) and standard deviation ($\Delta V_{zero}$) of the offset terms we used were slightly larger: 0.51V and 0.071V for the shallow site, 0.41V and 0.045V for the deep site (compared to 0.15 and 0.015); our calculated electronic noise ($\Delta V_{out}$) was smaller (0.4 mV vs. 3 mV in Varadharajan), so we defaulted to their value. The standard deviation between the multiple trap gas samples was used as the uncertainty in $[CH_4]$. This term was generally small compared to the uncertainty due to AFTvol error. The cumulative errors were propagated by summing in quadrature.

The whole-lake surveys provide an estimate of $F_{CH4}$ integrated across the entire reservoir surface area and a 95% confidence interval range (Beaulieu et al., 2016). Variance estimates calculated from GRTS incorporates spatial autocorrelation, if present, resulting in smaller uncertainty ranges than survey approaches that ignore spatial autocorrelation (Stevens and Olsen, 2003). The GRTS design and data reduction were executed in R using the spsurvey package (Kincaid et al., 2019). We propagated the cumulative uncertainties across 2017 and 2018 by taking the 95% confidence interval of each survey and summing them in

quadrature.

**2.9 Statistical and Quantitative Analysis**

We quantified the relationship between sediment temperature (sedT) and $F_{CH4}$ using Q10 and breakpoint analyses. The concept of an "ecological Q10" (DelSontro et al., 2016) follows from the physiological exponential relationship between metabolic

processes and temperature. In contrast to physiological Q10 values, ecological Q10, hereafter "ecoQ10" values are muddied by time-lags and competing rate enhancers and inhibitors (e.g. that temperature affects both methanogens and methanotrophs, Segers, 1998; Duc et al., 2010; Lofton et al., 2014). While the physiological Q10 value for methanogenesis converges around 4 (Yvon-Durocher et al., 2014), ecoQ10 values for methane fluxes have been reported to range from 1 – 35 (e.g. DelSontro et al., 2016; Wik et al., 2014; Duc et al., 2010), and is calculated using the equation:

$$ecoQ10 = 10^{10b} \quad (6)$$

where b is the slope of the regression between temperature and $F_{CH4}$.





We also used a two-dimensional Kolmogov-Smirnov test (2DKS, Garvey et al, 1998) to quantify the temperature breakpoint distinguishing winter conditions where $F_{CH4}$ is near zero and unrelated to temperature from warm weather conditions where $F_{CH4}$ is elevated are positively correlated with temperature. The 2DKS test is a non-parametric statistic that uses measures of disagreement to define the largest difference between cumulative distribution functions, that is, a threshold or breakpoint

(Lopes et al., 2008). We applied the 2DKS test to each of the continuous $F_{CH4}$ monitoring datasets: EC, shallow AFT, and deep AFT, each for 2017 and 2018 for a total of six 2DKS tests. This threshold temperature represents the temperature above which substantial $F_{CH4}$ activity occurs.

We looked at diurnal patterns on monthly and daily timescales. For the monthly timescales we binned 30-minute periods and took the median. For daily timescales we adapted the methods used by Podgrajsek et al. (2014) to quantify "strong" diurnal

patterns. For 24-hour periods with at least eight night-time and eight daytime non-gap filled 30-minute flux measurements, we compared the median of daytime $F_{CH4}$ to night-time $F_{CH4}$. The period was defined as having a strong diurnal pattern if 1) the larger median was greater than the smaller median by more than 50% of the value of the smaller median, and 2) the contiguous points in the 30-minute times series were smooth, i.e. more similar than points separated in time. We determined smoothness using visual inspection.

## 3. Results

### 3.1 Seasonal patterns in $F_{CH4}$

We observed a consistent pattern of elevated $F_{CH4}$ during the warm season and low-magnitude $F_{CH4}$ during the cold season between the two monitoring years (Fig. 2) per the EC results. In both monitoring years, the majority of cumulative total CH4

emissions (>85%) occurred in the five-months between May 1 and September 30, when air and sediment temperatures were warmer (Fig. 3 (a)), and latent heat fluxes were elevated (Fig. 3 (b)). The mean (± SD) warm season $F_{CH4}$ was 14.6 ± 12.4 mg $CH_4$ $m^{-2}$ $hr^{-1}$ in 2017 and 17.3 ± 14.5 mg $CH_4$ $m^{-2}$ $hr^{-1}$ in 2018. Lower magnitude mean fluxes in the warm season of 2017 relative to 2018 correspond to lower mean air temperature (airT), sediment temperature (sedT), and latent heat flux (LE) during warm season 2017 than 2018 (Fig. 3, (a) and (b)). Mean warm season values for airT were 21.0 and 22.9°C in 2017 and 2018,

respectively; for sedT were 22.0 and 24.8°C in 2017 and 2018, respectively; and for LE were 80.7 and 97.3 $Wm^{-2}$ in 2017 and 2018, respectively. Both quantitative analyses of the relationship between $F_{CH4}$ and sedT yielded statistically significant results (Table 2). We observed ecoQ10 values ranging from 6 to 35, with correlation coefficient values that indicate a strong relationship for all observation sets (Table 2, Fig. 4). In both 2017 and 2018, the shallow AFT (water depth ~ 1.3 m) had a higher sedT threshold per the 2DKS test than the deep AFT (water depth ~ 8 m, Table 2, Fig. 4). The warm season period

corresponds with the period of lake stratification (Fig. 3 (g)): the thermocline developed in early May in both years; turnover occurred on 30 September in 2017 (with a weak turnover event occurring between 8-24 September), and on 7 October in 2018 (see Fig. S3 for more detail). Although we observed elevated levels of hypolimnetic dissolved CH4 at the deep site in 2018 (Fig. S4), we did not observe a substantial increase in $F_{CH4}$ coincident with turnover.

The Bowen ratio (the ratio of sensible heat flux to latent heat flux, H:LE, the proportion of warming to evaporation in the

energy budget) observed at our site was low in the summer (~0.1 - 0.2) and increased in the winter (~0.4), similar to what has been reported in other flux studies over open water (Liu et al., 2012, Vesala et al., 2006). Although we observed seasonal and





interannual differences in the Brunt-Väisälä frequency (Fig. 3 f), this indicator of underwater turbulence did not correlate with $F_{CH4}$ (Fig. S5, S6), in contrast to the findings of other studies of water-atmosphere trace gas exchange (c.f. Webb et al., 2019; MacIntyre et al., 2010). During the winter months $F_{CH4}$ dropped by more than an order of magnitude to a baseline close to zero: between 1 Nov and 1 April $F_{CH4}$ was $0.60 \pm 0.69$ mg $CH_4$ m$^{-2}$ hr$^{-1}$. The surface of Acton Lake was frozen for several periods

during the 2017-2018 winter: 27 Dec 2017-10 Jan 2018; 13-21 Jan 2018; and 5-15 Feb 2018, during which $F_{CH4}$ was $0.08 \pm 0.46$ mg $CH_4$ m$^{-2}$ hr$^{-1}$.

The most substantial difference between the two monitoring years is the period of elevated emissions in late May to early June observed in 2018 but not 2017 (hereafter "spring burst"). We define the 2018 spring burst as the period from 24 May thru 4 June, where the daily average $F_{CH4}$ observed by EC was $\geq 25$ mg $CH_4$ m$^{-2}$ hr$^{-1}$. Maximum $F_{CH4}$ of 62.0 mg $CH_4$ m$^{-2}$ hr$^{-1}$ occurred

on 29 May 2018. While the 2017 EC monitoring does indicate a small burst in $F_{CH4}$ of 20.4 mg $CH_4$ m$^{-2}$ hr$^{-1}$ on 5 June, overall $F_{CH4}$ was much smaller: mean $F_{CH4}$ for 24 May - 4 June 2017 was $3.6 \pm 1.8$ mg $CH_4$ m$^{-2}$ hr$^{-1}$. This difference was likely due to differences in available carbon and sedT, driven by differences in precipitation patterns in the two years (Fig. 3 (a) and (c)), discussed further in Section 4.2.1. The cumulative $CH_4$ emission over the 2018 twelve-day spring burst period was 10.8 g $CH_4$ m$^{-2}$ which is 17% of the total 2018 emission, and which accounts for 59% of the difference in cumulative annual emissions

between 2017 and 2018.

The differences between the 2017 and 2018 monitoring years continues past the early summer. During 2017, $F_{CH4}$ increased to a maximum in late summer, then declined back to the winter baseline. In contrast, the 2018 summer and fall were characterized by episodic emission pulses and declines before tapering down to the winter baseline. These patterns are reflected in the AFT results for the co-located shallow site, although the pattern is dampened (Fig. 2 (b)).

We found that the $F_{CH4}$ observations at Acton Lake did not have a clear over-arching diurnal pattern when aggregated over monthly timescales, (Fig. S7), but out of the 168 days with adequate data coverage for diurnal analysis, 18.5% (31 days) displayed strong diurnal patterns: sixteen with elevated daytime emissions and fifteen with elevated nocturnal emissions. Very few of these strong diurnal pattern days were contiguous: there were only four instances of strong diurnal patterns persisting for two or more consecutive days. While we looked for evidence of synoptic patterns in $F_{CH4}$ due to changes in overlying

pressure from frontal systems (c.f. Liu et al., 2016), we did not see this during the study period.

### 3.2 Spatial patterns in $F_{CH4}$

The results from the six spatial surveys indicate a consistent spatial pattern in $F_{CH4}$ that differs from previous findings on $CH_4$ emissions from temperate, eutrophic reservoirs which has shown that the river – reservoir transition zone near the tributary inlets tends to be a hot spot for emissions compared to the lacustrine zone (Beaulieu et al., 2014; Beaulieu et al., 2016; DelSontro

et al., 2011; Tuser et al., 2017). The survey results from Acton Lake indicate relatively consistent rates of $F_{CH4}$ across most of the reservoir surface area (Fig. 5), and a weak but significant (n=90, $R^2 = 0.1$, $p < 0.005$) positive relationship between ebullition and reservoir depth (Fig. 6).

At the whole-lake scale, ebullition was a dominant emission pathway for $CH_4$ relative to diffusion, accounting for 82-94% of total $F_{CH4}$. However, at certain sites diffusive $F_{CH4}$ contributed a larger proportion of the total flux (Fig. 7). The four sites with

mean ebullitive to total $F_{CH4}$ ratios less than 0.8 are also the four shallowest sites (see Fig. 1): U-09, U-14, U-07, and U-06, with mean observed depths of 1, 1.3, 1.5, and 2 m respectively. Emission behaviour at sites U-09 and U-06 was substantially





different than at other sites: these two sites had consistently low $F_{CH4}$ (Fig. 5) and tend to have higher rates of $CH_4$ diffusion than ebullition. Much of this behaviour is likely explained by the proximity of these sites to Acton Lake's swimming beach, which has a sandy substrate. This difference in sediment composition likely inhibits methanogenesis at these sites.

### 3.3 EC gap filling and uncertainty

The overall $F_{CH4}$ data acceptance rate for the two-year monitoring period (26 January 2017 – 13 November 2018) was 31.3% (Fig. S8). In 2017, the data acceptance rate was lower, 23.4%, due to power issues and the need to filter for wind direction at the near shore EC S-1 site where the instrumentation was located for the whole year vs. 39.8% in 2018 when the instrumentation was relocated in the spring to the mid-lake EC S-2 site. The data coverage for the period of monitoring from EC S-2 (May thru November) was 52.8%. Re-siting removed the need to filter periods based on wind direction and coincided with an

improvement to the battery system that reduced incidences of power failure. At EC S-1, non-operability of the LI7700 due to power loss or other issues caused the majority of data rejection (40.4% of total monitoring periods), followed by filtering for wind direction (28.1%), and quality control filtering (7.8%). At EC S-2, power loss caused the majority of gaps (36.3%), followed by quality control filtering (16.6%).

The non-gap filled, quality filtered 30-minute $F_{CH4}$ measurements had a mean random error (± SD) of 1.3 ± 1.9 and 1.8 ± 1.7

mg $CH_4$ m$^{-2}$ hr$^{-1}$ in 2017 and 2018, respectively or 15.5% and 13.7% of the mean annual fluxes. The fractional errors were larger in the winter months when $F_{CH4}$ was small (mean winter random error: 23%) and smaller during the warmer months when $F_{CH4}$ was larger (mean summer random error: 15%). Both the magnitudes and patterns in the random errors are similar to those observed by Jammet et al. (2017) in a subarctic aquatic ecosystem. Similarly, we found gap-filling our $F_{CH4}$ time series with ANN worked well with a few exceptions. The median $R^2$ value for the 20 extractions was 0.79, and the cumulative bias

error was minimal: the 20 ANN extractions yielded a median bias of 0.25 (range of -3.7 to 3.5) g $_{CH4}$ m$^{-2}$, or up to 3.3% of cumulative emissions over the two-year monitoring period. The ANN establishes non-linear predictive power to each of the driver inputs, defined as a "Variable Importance Factor" (VIF) in terms of a percent importance to the predictive power of the model. The median VIFs from the 20 ANN extractions are plotted in Fig 8; a consistently high ranking across runs indicates a strong relationship with $F_{CH4}$.

### 3.4 $F_{CH4}$ warm season and annual budgets

The larger-magnitude $F_{CH4}$ emissions observed in 2018 relative to 2017 at Acton Lake carries through to the warm-season and annual budgets across measurement methods. The EC and GRTS survey results indicated similar warm-season mean fluxes in 2017: 9.69 ± 0.67 and 9.98 ± 6.2 mg $CH_4$ m$^{-2}$ hr$^{-1}$ (Table 2). Results from both methods indicated larger-magnitude mean $F_{CH4}$ in 2018: 17.45 ± 0.38 mg $CH_4$ m$^{-2}$ hr$^{-1}$ per the EC system and 13.01 ± 6.6 mg $CH_4$ m$^{-2}$ hr$^{-1}$ per the GRTS surveys (Table 2).

Both the shallow-site and deep-site results also indicated elevated $F_{CH4}$ in 2018 relative to 2017 (Table 2). The lower-magnitude mean $F_{CH4}$ measured by these methods relative to EC measurements is likely due to the under-representation of hot-spots (Wik et al., 2016).

Approximately 87.5% of cumulative annual $F_{CH4}$ occurred during the five-month warm-season period from May thru September in 2017, and ~89.7% in 2018 per the EC results. The cumulative areal emissions during the warm-season in 2017

were 35 ± 2.5 g m$^{-2}$ (Fig. 9 (a)), while the annual cumulative areal emissions in 2017 were 40.6 ± 5.9 g $CH_4$ m$^{-2}$. Similarly, in



2018 the cumulative areal emissions during the warm-season were $64.1 \pm 1.4$ g m$^{-2}$ (Fig. 9 (b)), compared to the annual rate of $71.4 \pm 4.2$ g CH$_4$ m$^{-2}$. Scaling up to the 2.4 km$^2$ area of Acton Lake, these measurements indicate that this reservoir was a source of $97.4 \pm 14$ and $171 \pm 10$ Mg CH$_4$ to the atmosphere in 2017 and 2018, respectively.

## 4. Discussion

**4.1 Biophysical drivers**

We intensively monitored one small eutrophic reservoir, Acton Lake, over two years in this study with the goal of improving our understanding of the patterns and drivers of F$_{CH4}$ from reservoirs. The drivers we used to run the ANN to gap-fill the EC time series (Section 2.7) were chosen because of their known biophysical links to either CH$_4$ production (sediment temperature and air temperature) or emission across the air-water interface (change in static pressure, absolute static pressure, and latent

heat flux). We also included empirical factors that do not fall into either category: a location indicator (Site 1 vs. Site 2) and indicators for day of year and hour of day. Day of year (DOY) was the top predictor, likely reflecting the strong seasonal pattern in F$_{CH4}$ (Fig. 8). That "Site", an indicator of where the flux tower was located, came out as the second most important input is likely reflective of the difference between years. While the re-siting of the tower (even given the mitigating factor of overlapping footprints) does make it difficult to disentangle some spatial and temporal differences, the spring burst is a clear

signal we did not see in 2017 and constitutes the majority of the difference between years. In terms of biophysical drivers, sediment temperature, absolute static pressure, change in static pressure, and latent heat flux were ranked as the top drivers (Fig. 8). We used ecoQ10 and 2DKS threshold analysis to further investigate the role of sediment temperature on regulating F$_{CH4}$. Static pressure and latent heat flux are important drivers of diurnal patterns in F$_{CH4}$ (Section 4.2.2.).

The ecoQ10 values and 2DKS threshold temperatures were consistent across years among the three continuous monitoring

methods (Table 2). The EC method had the lowest ecoQ10 value (~6), indicating the weakest relationship between sediment T and observed emissions. This may be due to two factors: the first is the spatial decoupling of the EC measurements from the sedT thermistor, as the EC measurement footprint encompasses a wider swath of the reservoir, much of which may have different sedT conditions. The second factor is that the EC measurements integrate ebullitive and diffusive emissions. While diffusive emissions are dependent on methanogenesis and thus sedT, the relationship may be confounded by methanotrophy,

which is temperature dependent as well (Fuchs et al., 2016). The shallow site AFT had a consistently higher ecoQ10 value than the deep AFT site, indicating a stronger relationship between sedT and ebullitive F$_{CH4}$ at the shallow site than the deep site. Despite a greater ecoQ10 value, ebullitive F$_{CH4}$ at the shallow site didn't respond to warming in the spring until water temperatures reached ~22.5 °C, whereas ebullitive F$_{CH4}$ at the deep site responded to warming at a much lower temperature threshold (13 – 18 °C). Furthermore, mean ebullitive F$_{CH4}$ was very similar between the two sites (Table 3), despite a 6-degree

difference in maximum sediment temperature. These patterns suggest that methanogens at the deep site may be better adapted to the consistently cooler conditions found in the hypolimnion of Acton Lake, which has important implications for predictive models employing ecoQ10 or threshold values to parameterize F$_{CH4}$ as a function of sedT. Alternatively, the differences in temperature sensitivity between the deep and shallow site may reflect differences in substrate quality and/or quantity related to spatial patterns in sedimentation and productivity (Berberich et al. 2019). Regardless of the underlying mechanism, these

patterns illustrate strong spatial patterning in CH$_4$ biogeochemistry within this 2.4 km$^2$ reservoir.





Another indication of the intra-lake variability is the time-lag between the shallow and deep site in both maximum sedT and maximum $F_{CH4}$. Maximum ebullitive $F_{CH4}$ observed by the AFTs coincided with maximum sedT at both the shallow (U-14) and deep (U-12) monitoring sites in 2017 (Fig. 10). This maximum occurs in early August at U-14 versus mid-September at U-12, a phase-shift that reflects the time delay in heat transfer to the deeper sediment, and nutrient transfer due to the distance from the inlets. This pattern was not as pronounced in 2018 (Fig. S9), likely because other drivers played a larger role in $F_{CH4}$ that season, in line with the results of the ecoQ10 analysis indicating a stronger relationship between $F_{CH4}$ and sedT in 2017 than in 2018 (Table 3, Fig. 4).

### 4.2 Temporal Patterns

#### 4.2.1: Interannual & Spring Burst

Our analysis suggest that precipitation dynamics drove differences in algal populations that lead to both the 2018 spring burst and the subsequent differences in warm season emission patterns between years. The spring burst period of elevated $F_{CH4}$ in late May thru early June of 2018 indicates that environmental conditions were more favourable for $CH_4$ production and emission in the late spring of 2018 compared to the same period in 2017. Spring of 2017 was relatively wet, with 31.0 cm of rainfall and 20.9 *$10^6$ $m^3$ of stream inflow in May (Fig. 3 (c), (d)) which drove substantial fluctuations in reservoir water level (Fig. 3 (e)). These rain events also led to a decrease in sedT from 22.5 to 18°C prior to the onset of the spring burst timeframe (Fig. 11 (a)) due to the inflow of cooler stream water and the cooling of ambient air temperature. In contrast, May of 2018 was relatively dry, with 12.3 cm of rain, 9.45 *$10^6$ $m^3$ of stream inflow (Fig. 3 (c), (d)), and stable reservoir water levels (Fig. 3 (e)). Prior to the spring burst in 2018, sedT rose from 22.5 to 27°C (Fig. 11 (a)). Chlorophyll levels in spring 2018 were elevated compared to spring 2017 (Fig. 11 (b)). This suggests enhanced production of algal biomass, presumably due to differences in precipitation, turbidity, and water level stability.

Precipitation and stream inflows have been shown to have contrasting effects on reservoir carbon dynamics. An increase in precipitation can increase allochthonous carbon (C) availability in the reservoir due to inflow of dissolved and particulate C in the stream discharge, but enhanced turbidity from stream discharge can decrease autochthonous C availability due to shading. Another study at Acton Lake found that storm events led to suppression of lake metabolism, likely due to the flushing removal of autotrophic biomass (Williamson et al., 2020). Thus, precipitation can shift the balance between autochthonous C (autoOC) and allocthonous C (alloOC), and the source of C can impact $CH_4$ production and emission. Several lab studies have found that additions of autochthonous C (autoOC), such as algal cells, stimulate $CH_4$ production rates (Schwarz et al. 2008; West et al. 2012, 2015), and West et al., (2012) found that additions of autoOC lead to a greater increase in $CH_4$ production than additions of allochthonous C (alloOC); Grasset et al., (2018) found that autoOC decomposed more rapidly than alloOC, and that additions of the former lead to pulses of $F_{CH4}$, while additions of the latter lead to slower and more constant $CH_4$ production and emission. This would be consistent with the emission patterns in 2018 and 2017, respectively: warm temperatures and low turbidity in May 2018 lead to an algal bloom and high levels of autoOC resulting in the spring burst of $F_{CH4}$, while a wet spring in 2017



lead to cool, turbid inflow that suppressed an algal bloom, but loaded the reservoir with slow-burning alloOC. Hence in 2017 the steady increase in $F_{CH4}$ that tracked with sedT and accumulating algal biomass to a maximum in late fall.

### 4.2.2: Diurnal patterns

Diurnal patterns in $F_{CH4}$ were less prevalent at Acton Lake than in some other systems reported on in the literature. Understanding the amplitude and direction of any diurnal pattern in $F_{CH4}$ helps to reduce bias in upscaled survey-style measurements of $F_{CH4}$, which for convenience and safety are almost always collected during the daytime. It is noteworthy that Acton Lake could switch diurnal pattern phases, which we propose is due to shifts in biophysical controls on $CH_4$ production and emission. We only observed strong diurnal patterns 18.5% of the time, compared to reports of consistent diurnal patterns

observed in other eddy covariance studies of $F_{CH4}$ in a nutrient rich Swedish lake (Podgrajsek et al., 2014), a mesotrophic subtropical reservoir (Deshmukh et al., 2014), and a subarctic pond (Jammet et al., 2017). The lack of a diurnal pattern >80% of the time and the lack of a relationship between $F_{CH4}$ and underwater turbulence (Fig. 3 (f), S5, S6) indicate factors that do not vary diurnally are important in controlling $CH_4$ production and emission at Acton Lake and perhaps in warm, eutrophic reservoirs more broadly. While sedT, LE, and static pressure all contribute to the regulation of $F_{CH4,}$ the roles of substrate

availability and microbial community dynamics, which may not vary on a diel basis, may play a larger role than the physical factors at Acton in regulating $F_{CH4}$.

Although periods of contiguous diurnal patterns were relatively rare at Acton Lake, they co-occurred with diurnal patterns in potential drivers. Contiguous daytime peaks in $F_{CH4}$ correlated with peaks in latent heat flux (LE), a measure of the heat energy used in evaporation of liquid water from the surface (Fig. 12 (a), (c); Fig. S10). This is consistent with the ANN results which

identified LE as an important predictor variable (Fig. 8). While LE is not a direct biophysical driver of $CH_4$ production or emission, it is enhanced under warm, windy, convective conditions (Liu et al., 2012). These same conditions may enhance diffusive $CH_4$ emissions by 1) causing deeper, $CH_4$-rich water to upwell toward the water surface, and 2) enhancing the air-water gas exchange rate. Periods of night-time $F_{CH4}$ peaks (Fig. 12 (b), (d)) co-occurred with night-time minimums in atmospheric pressure, consistent with static pressure serving as an important predictor in the ANN (Fig. 8, Fig. S11). Drops in

overlying static pressure can trigger the release of sediment-bubbles (Beaulieu et al., 2018; Harrison et al., 2017; Varadharajan and Hemond, 2012), thereby greatly enhancing ebullitive $F_{CH4}$. A similar result was reported for a tropical reservoir, but in that case atmospheric pressure was typically lower during the day than at night, resulting in consistent daytime peaks in $F_{CH4}$ (Deshmukh et al., 2014). Diurnal patters in static P do not necessarily result in a corresponding pattering $F_{CH4}$ at Acton Lake, however, likely because other factors also exert control on $F_{CH4}$. The period shown in Fig. 12 b, d had high, stable sedT (not

shown) which may have minimized the role of temperature as a limiting factor, enhancing the role of static P.

### 4.3 Comparison with other systems and methods

The cumulative annual $CH_4$ areal emissions were 40.6 ± 5.9 g $CH_4$ m$^{-2}$ in 2017 and 71.4 ± 4.2 g $CH_4$ m$^{-2}$ in 2018 at Acton Lake, based on the gap-filled continuous EC monitoring results (Fig. 9). Comparing these values to others reported in the

literature is not straightforward, due in large part to the temporal coverage: the majority of studies that report $F_{CH4}$ from inland



waters monitor during the warm season, with less than six months of measurements (cf. Deemer et al., 2016; DelSontro et al., 2018 Bastviken et al., 2011). Lack of shoulder-season and winter measurements could bias the mean or cumulative $F_{CH4}$ high; lack of continuous measurements that miss hot-moments could bias mean/cumulative $F_{CH4}$ low.

Upscaling survey-style methods is analogous to gap-filling pseudo-continuous methods like EC, but there is currently a
disconnect in terms of how the uncertainty in the two processes is characterized. Within the flux community, ANN methods for gap-filling $F_{CH4}$ time series are being adopted, but it is an area of ongoing research (Nemitz et al., 2018; Knox et al 2019). While several studies over wetlands and rice have used ANN to gap-fill $F_{CH4}$ (e.g. Dengel et al., 2013; Morin et al., 2014; Knox et al., 2015, Rey-Sanchez et al., 2017; Knox et al., 2016), to our knowledge, Jammet et al. (2017) is the only other study that has gap-filled open-water $F_{CH4}$ measurements using ANN. Gap-filling with ANN gives non-linear weighted best estimates of
$F_{CH4}$ in relation to the drivers for that time period. The ANN also enables us to characterize uncertainty by looking at the upper and lower confidence intervals of the ensemble models. This gives us a continuous time series of uncertainty integrated over the footprint of the EC system. In contrast, the uncertainty of survey-style methods often derives from spatial heterogeneity. The GRTS surveys are characterizing the lake-scale $F_{CH4}$ (and uncertainty) as a snapshot in time. Thus, upscaling these survey results temporally is a matter of gap-filling by linear interpolation.

While these methodological differences are noted, nevertheless Acton Lake's annual $F_{CH4}$ is relatively high. It falls in the 4th quintile (>60%) of the reservoir emission rates that included ebullition reported in Deemer et al., (2016); the warm season $F_{CH4}$ fall in the upper quintile (>80%) of those reservoirs. Deemer et al., (2016) also discuss how methods matter in characterizing $F_{CH4}$ from reservoirs. The main consideration is the importance of measuring ebullition; Deemer at al. found that the mean $F_{CH4}$ reported in studies measuring ebullition and diffusion was over double of diffusion-only $F_{CH4}$ studies. A secondary
consideration is a method's ability to capture spatial and temporal variability of $F_{CH4}$. Deemer et al. noted that studies using the eddy covariance method reported substantially higher values of mean $F_{CH}$: ~190 mg $CH_4$-C $m^{-2}$ $d^{-1}$ (Deshmukh et al., 2014) and ~328 mg $CH_4$-C m-2 d-1 (Eugster et al., 2011), which are comparable to the Acton Lake annual $F_{CH4}$ values converted to daily mean fluxes: 83.4 and 147 mg $CH_4$-C $m^{-2}$ $d^{-1}$ in 2017 and 2018, respectively. The two open-water sites included in the Knox et al., (2019) analysis had mean annual $F_{CH4}$ of ~15 g $CH_4$ m-2 y-1, or 5.5 mg $CH_4$ $m^{-2}$ $d^{-1}$. These sites are natural lakes
in temperate regions. This difference in $F_{CH4}$ speaks to the need for building a representative dataset across both methods and ecoregions.

The difference in $F_{CH4}$ between the different methods in 2017 versus 2018 (Table 2) illustrates how methods matter in capturing hot-spots and hot-moments. In 2017, the warm-season $F_{CH4}$ measured by EC and the GRTS surveys are not significantly different, and $F_{CH4}$ measured at the two AFT sites is within 30% of this value. This suggests that the GRTS survey adequately
captured hot spots of emissions and did not under-represent hot moments. In 2018, by contrast, the warm-season $F_{CH4}$ measured by EC is 34% larger than the GRTS survey results, and more than double what was measured at the AFT sites. Using a literature model to estimate $F_{CH4}$ also yields estimates that agree relatively well with 2017 EC results, but not 2018: 11.1 and 10.3 mg $CH_4$ $m^{-2}$ $hr^{-1}$, respectively (DelSontro et al., 2018). This reflects the limitations in the datasets used to train the model: the model uses chla levels to predict $F_{CH4}$. Mean chla levels were higher in 2017 than 2018 for Acton Lake, but as we saw here, the timing
of the elevated chla levels was key in how that input translated to $F_{CH4}$. The difference in uncertainty between the methods should also be noted. While we report a larger uncertainty range for the GRTS survey results than either the EC results or the AFTs and chambers (Table 2, Fig. 9), the GRTS uncertainty is the only one that encompasses the whole lake.



Integrating spatially over the 2.4 km$^2$ surface area of the lake, these results suggest Acton Lake emits $111 - 155$ Mg CH$_4$ yr$^{-1}$. This loss of mineral carbon (C) is non-trivial term in the reservoir C budget, constituting ~7% (range: 4%-12%) of Acton Lake's annual C retention over 2007 and 2008 reported in Knoll et al. (2013). The Knoll study did not measure $F_{CH4}$ and thus did not include it in their budget. They did include losses of C as CO$_2$, which tended to be much smaller in magnitude than

CH$_4$-C emissions: 2 and 56 Mg C yr$^{-1}$ in 2007 and 2008, respectively, compared to 83 and 116 Mg C yr$^{-1}$.

## 5.   Conclusions

In this study we investigated temporal patterns and biophysical drivers of CH$_4$ fluxes from a eutrophic temperate reservoir using multiple methods including eddy covariance. Sediment temperature and the overlying static pressure were the most

important biophysical drivers of $F_{CH4}$. Precipitation patterns were found to be important because of how inflow affected both sediment temperature and algal biomass dynamics. In contrast to previous studies, we did not find a strong relationship between $F_{CH4}$ and underwater turbulence.

We found that the study system, Acton Lake, had cumulative annual CH$_4$ areal emissions of $40.6 \pm 5.9$ and $71.4 \pm 4.2$ g CH$_4$ m$^{-2}$ in 2017 and 2018, respectively. These levels of emissions place Acton Lake in the upper quartile of emission rates reported

from reservoirs (Deemer et al., 2016), further supporting the concept that system productivity is a more important factor than latitude in predicting CH$_4$ emission rates (Del Sontro et al., 2018). A spring burst of $F_{CH4}$ observed over a two-week period in 2018 but not 2017 accounted for 59% of the difference in cumulative emissions between years. This difference between consecutive years highlights the importance of multi-year studies (c.f. Room et al., 2014), and the importance of characterizing temporal variability in open water systems, which Williamson et al. (2020) illustrated exceeded spatial variability for several

physical, chemical, and biological metrics.

The EC technique holds much promise for improving our understanding of the biophysical drivers of gaseous fluxes, with a few caveats. In addition to the pseudo-continuous temporal coverage, the EC measurement footprint encompasses a much larger area than traditional gas flux measurement techniques (e.g. dissolved gas sampling, chambers, inverted funnel traps), increasing the likelihood of integrating fluxes over a distribution of hot spots. However, care must be taken in the siting, quality

control, and interpretation of results. The authors reemphasize the recommendation given by Vesala et al., (2012): for best results, close collaboration is needed between biometeorologists and limnologists, to understand what is going on both above and below the water.

The EC results in this study further our understanding of the interaction between precipitation, sediment temperature, algal productivity levels, and $F_{CH4}$. This study adds to our understanding of open water flux processes at appropriate spatial and

temporal scales, while highlighting a way to present and compare EC and whole-lake survey data in appropriate contexts.



**Code and Data Availability**

The datasets and R code used for the analysis in this study are available on Zenodo. The raw data and R code are available under: R Code for: Temporal patterns and biophysical controls on methane emissions from a small eutrophic reservoir: insights from two years of eddy covariance monitoring, doi: 10.5281/zenodo.4540271; and supplemental ANN resampling data is

available under: Artificial Neural Network (ANN) Resampling Results for Gap Filling Eddy Covariance Data, doi: 10.5281/zenodo.4540271.

**Author Contributions**

1.  S. Waldo: conceptualization, data curation, formal analysis, investigation, methodology, project administration, software, visualization, writing – original draft, writing – review & editing

2.  J.J. Beaulieu: conceptualization, data curation, formal analysis, funding acquisition, investigation, methodology, project administration, resources, software, supervision, writing – review & editing

3.  W. Barnett: formal analysis, methodology, software, writing – review & editing

4.  D.A. Balz: conceptualization, data curation, investigation, methodology, project administration, resources, supervision

5.  M.J. Vanni: data curation, formal analysis, investigation, resources, writing – review & editing

6.  T. Williamson: data curation, formal analysis, investigation, resources

7.  J.T. Walker: conceptualization, funding acquisition, investigation, methodology, project administration, resources, supervision, writing – review & editing

**Disclaimer**

The views expressed in this article are those of the authors and do not necessarily reflect the views and policies of the US

Environmental Protection Agency. Any mention of trade names, manufacturers or products does not imply an endorsement by the United States Government or the US Environmental Protection Agency. EPA and its employees do not endorse any commercial products, services, or enterprises.

**Acknowledgements**

We thank David Wesler and other personnel at Hueston Wood State Park for all of their support in our monitoring efforts at

Acton Lake. We are very grateful to the members of the EPA Scientific Dive Unit for their assistance in installing the mid-lake tower: Steve Donahue, Brad White, Frank Borsuk, David Light, Nathan Doyle, and Leah Ettema. We also thank Gil Bohrer



and Jorge Villa for their guidance and assistance with the mid-lake tower. We thank Ryan Daly, Bill Mitchell, and Garrett Wiley for assistance with design and fabrication of tower hardware and power systems. We are grateful for the additional laboratory and field support provided by Karen White, Paul Trygstad, Eleanor Silver, Megan Berberich, Keith Bisbe, Aiden Pemberton, Page Jordan, and Tom Radford. We acknowledge that Acton Lake is located within the traditional homelands of

5  the Myaamia and Shawnee people, who along with other indigenous groups ceded these lands to the United States in the first Treaty of Greenville in 1795.





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





**Table 1: Measurement methods summary**

| Method | Flux Measured | Spatial Coverage | Frequency | Use |
|---|---|---|---|---|
| Eddy Covariance (EC) | total net | ~100s m$^2$, north sector of the lake | pseudo-continuous, 30-min timestep | • annual budgets<br>• diurnal patterns<br>• biophysical drivers: ANN, Q10, 2DKS |
| Active Funnel Traps (AFT) | ebullition | 0.3 m$^2$, two locations | pseudo-continuous, 30-min timestep | • annual budgets<br>• diurnal patterns<br>• biophysical drivers: Q10, 2DKS |
| Flux Chamber | diffusion | 0.2 m$^2$ per site | 2 sites bi-weekly; 15 sites sampled during 6 GRTS surveys | • annual budgets<br>• emission pathway relative importance |
| Passive Funnel Traps | ebullition | 0.3 m$^2$ | 15 sites sampled during 6 GRTS surveys | • annual budgets<br>• spatial patterns<br>• emission pathway relative importance |



**Table 2: Summary statistics describing the relationship between $F_{CH4}$ and sediment temperature per the ecoQ10 analysis and the two-dimentional Kolmogov-Smirnov test (2DKS) threshold analysis**

|  |  | Eddy Covariance | AFT Shallow | AFT Deep |  |
| --- | --- | --- | --- | --- | --- |
| ecoQ10 | 2017 value | 6.96 | 35.1 | 30.4 | 5 |
|  | 2017 $R^2$ | 0.85 | 0.48 | 0.60 |  |
|  | 2018 value | 5.64 | 35.8 | 30.7 |  |
|  | 2018 $R^2$ | 0.83 | 0.85 | 0.38 |  |
| Threshold (2DKS) | 2017 sedT threshold | 14.1 | 22.2 | 17.9 |  |
|  | 2017 test statistic | 0.226 | 0.166 | 0.204 | 10 |
|  | 2018 sedT threshold | 17.4 | 23.0 | 13.3 |  |
|  | 2018 test statistic | 0.234 | 0.190 | 0.138 |  |

**Table 3: Seasonal methane fluxes reported as mean fluxes and cumulative areal emissions from Acton Lake characterized by different measurement techniques. The eddy covariance method measures total (diffusive + ebullitive + other) fluxes.**

|  | Warm Season[1] 2017 Mean Flux (mg CH$_4$ m$^{-2}$ hr$^{-1}$) | | | Warm Season[1] 2018 Mean Flux (mg CH$_4$ m$^{-2}$ hr$^{-1}$) | | |
| --- | --- | --- | --- | --- | --- | --- |
|  | **Diffusive** | **Ebullitive** | **Total** | **Diffusive** | **Ebullitive** | **Total** |
| Eddy Covariance | -- | -- | 9.69 ± 0.67 | -- | -- | 17.45 ± 0.38 |
| U-14, Shallow Site | 3.19 | 4.38 ± 0.63 | 7.58 ± 0.63 | 2.47 | 5.66 ± 0.11 | 8.13 ± 0.11 |
| U-12, Deep Site | 0.89 | 6.35 ± 0.54 | 7.24 ± 0.54 | 1.23 | 6.23 ± 0.05 | 7.46 ± 0.05 |
| GRTS Surveys | 1.28 ± 0.52 | 8.71 ± 6.1 | 9.98 ± 6.2 | 1.87 ± 1.2 | 11.13 ± 6.1 | 13.01 ± 6.6 |

---

[1]"Warm Season" is defined as 1 May - 30 September

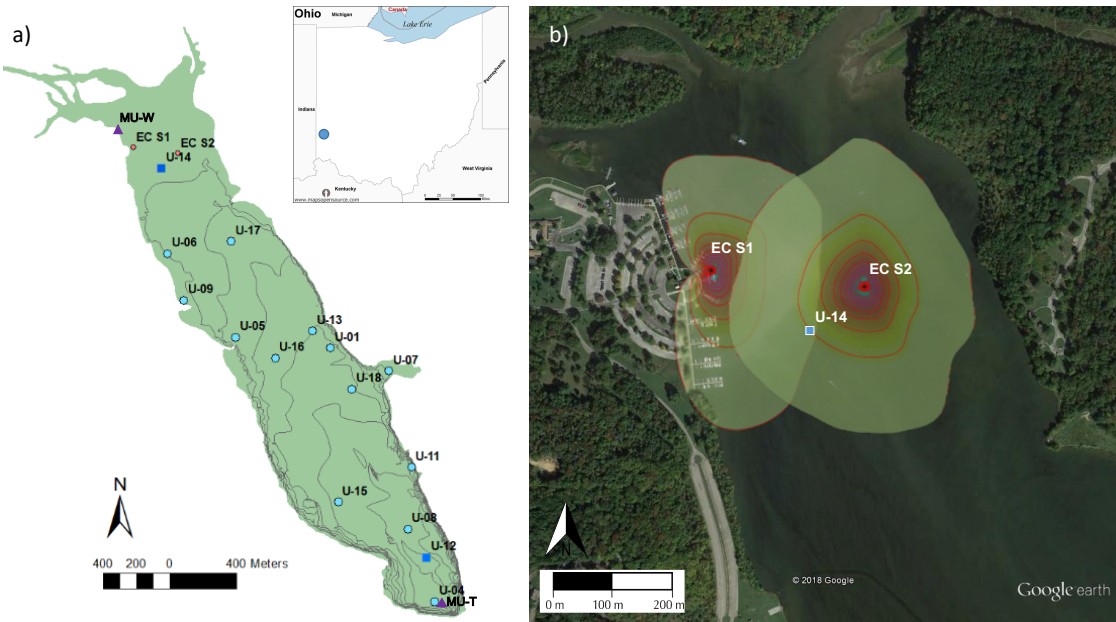

**Figure 1: Map of Acton Lake (a), showing the location of multiple monitoring methods: eddy covariance flux tower sites (red circles), active funnel traps and bi-weekly chamber measurements (dark blue squares), and spatially extensive survey sites (light blue circles), and the weather station and thermistors operated by Miami University (purple triangles). The lake contour lines represent ~ 1m depth increments. Inset image shows the location of Acton Lake in southwest Ohio. The © Google Earth image (b) shows the 80% cumulative footprint probability distribution at each eddy covariance flux tower site at 10% intervals**



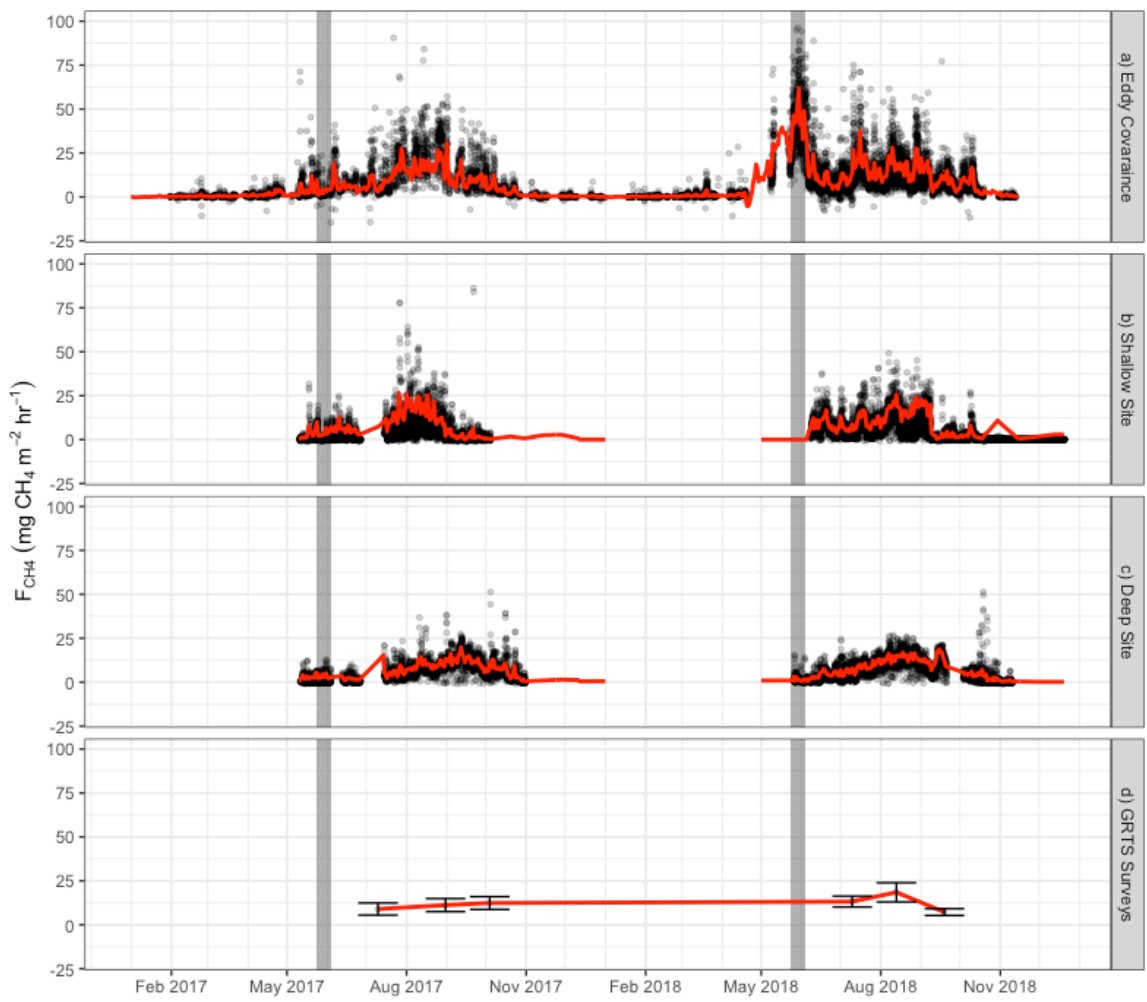

**Figure 2: Time series of F$_{CH4}$ monitored via multiple methods: eddy covariance (panel a), the shallow AFT (panel b, site U-14), and the deep AFT (panel c, site U-12), and via the spatially integrated lake-wide surveys (panel d). Black circles are observed fluxes, and red traces show the daily mean (sum of ebullition and interpolated chamber diffusion measurements for panels b) and c). The error bars in panel d) indicate the 95% confidence interval of the mean. Vertical grey bars indicate the 2018 spring burst period, 24 May - 4 June.**



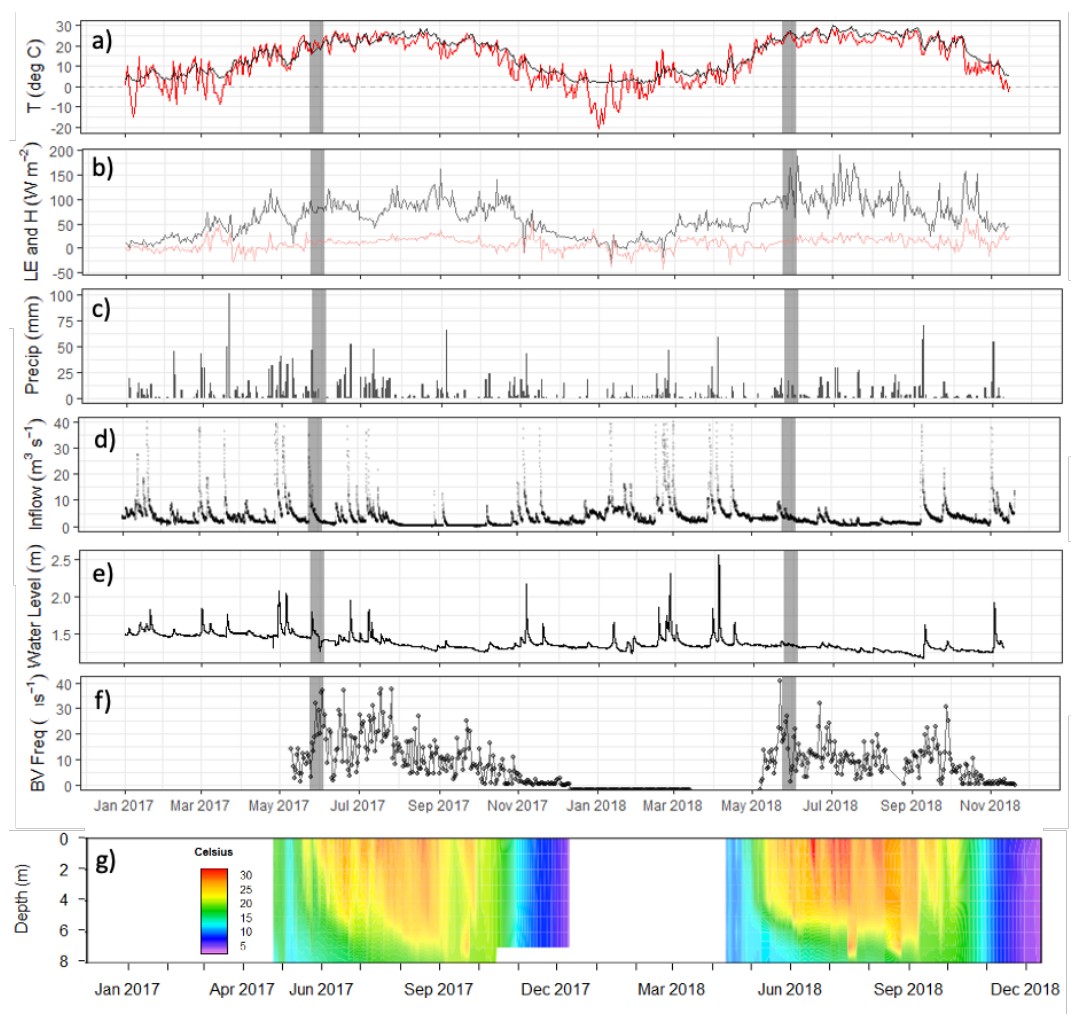

**Figure 3: Meteorological and limnological conditions over the study period: (a) daily mean of air (red) and sediment (black) temperature; (b) daily mean latent and sensible heat fluxes (LE, black, and H, red, respectively); (c) daily cumulative precipitation (mm); (d) stream inflow (m³ s⁻¹); (e) water depth in the footprint of the flux tower (m); (f) Brunt Väisälä frequency, a measure of water column mixing potential (s⁻¹); and (g) the water temperature profile at the deep site (U-12). Grey bars indicate the time frame of the 2018 spring burst of CH₄ emissions.**

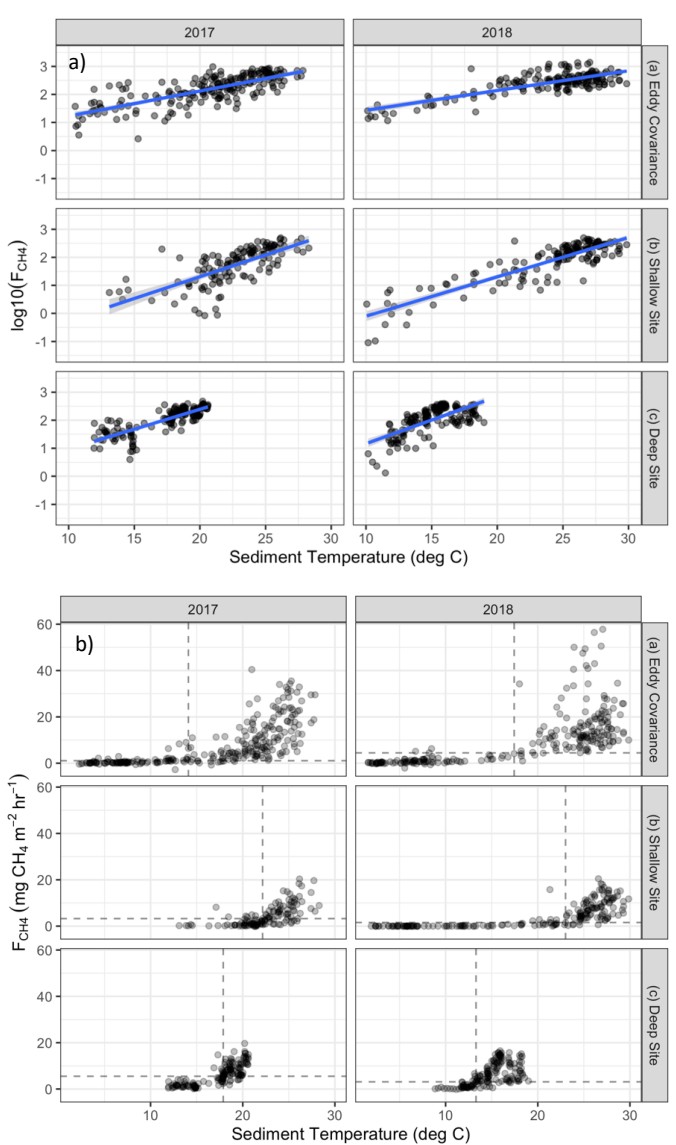

**Figure 4:** $F_{CH4}$ **measured by EC (top), the shallow AFT (middle) and the deep AFT (bottom) in 2017 and 2018 as a function of sedT. The panel (a) shows linear regressions of log-transformed $F_{CH4}$ vs sedT to determine the ecoQ10 (Table 3). The panel (b) show the threshold values for $F_{CH4}$ as a function of sedT: the dotted lines are the x- and y-breakpoints determined via two-dimensional Kolmogov-Smirnov tests.**

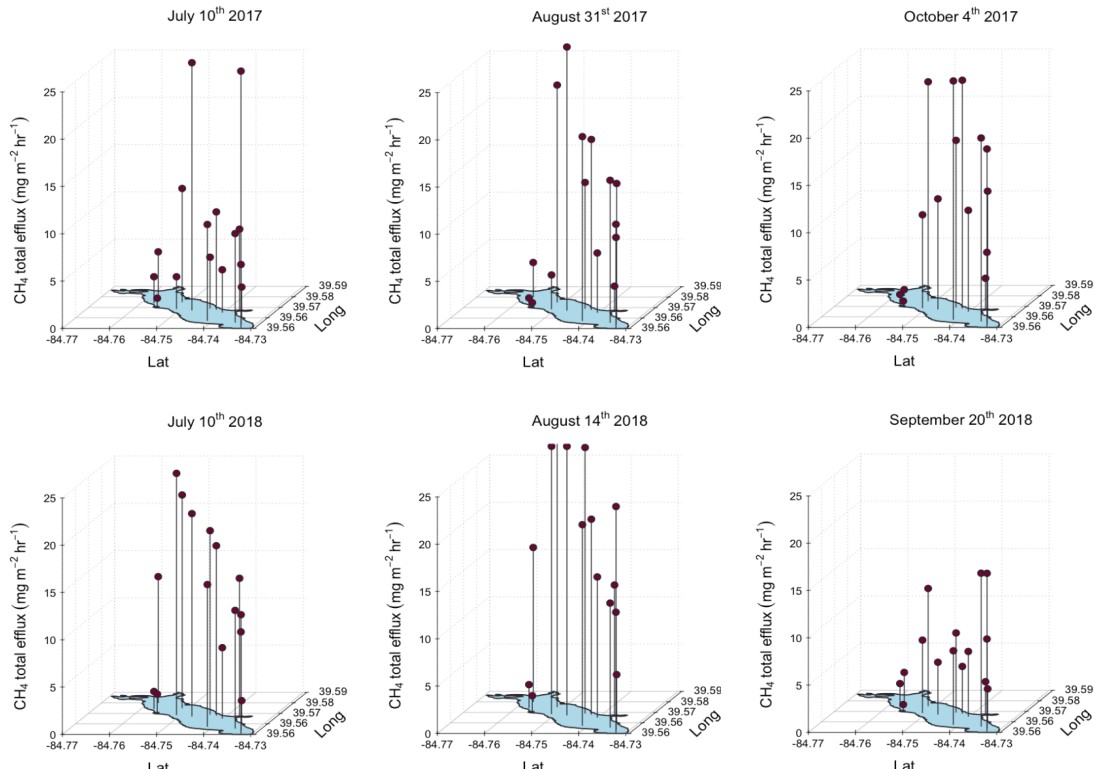

**Figure 5: Total (ebullitive + diffusive) $F_{CH4}$ measured during mid-summer, late-summer, and fall spatial surveys at Acton Lake during 2017 (top row) and 2018 (bottom row). Dots indicate magnitude of $F_{CH4}$ per the z-axis scale and vertical black lines connect red dots to their corresponding sampling location.**



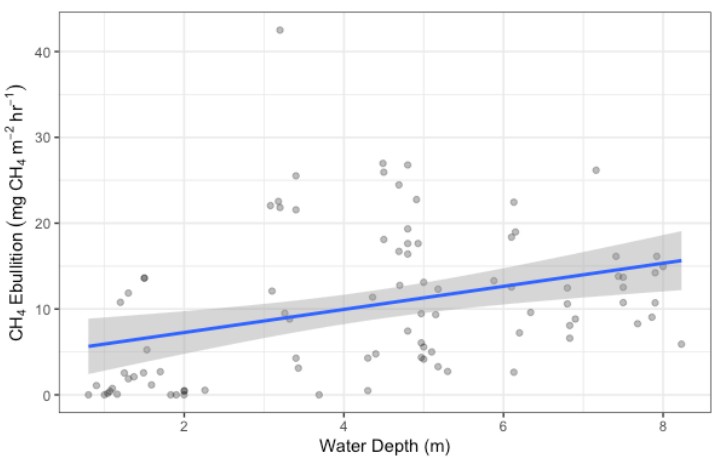

**Figure 6: Linear regression of the spatial survey CH₄ ebullition results (n=90, measured with PFTs) as a function of water depth at the site. $R^2 = 0.1$, $p<0.005$.**

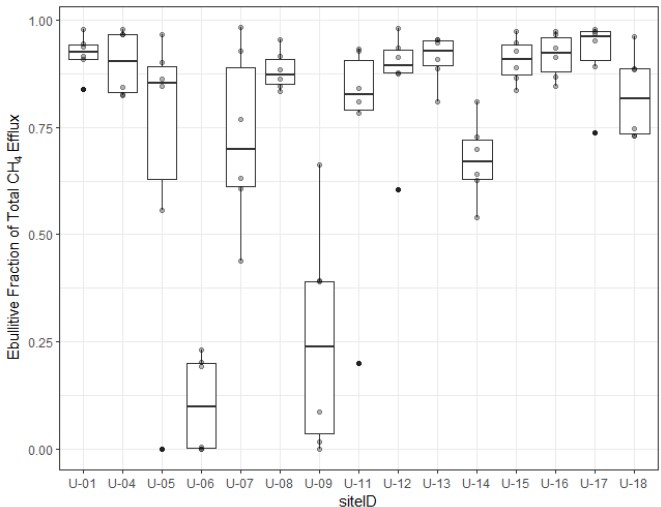

5    **Figure 7: Mean, interquartile range, and standard deviation of the fraction of ebullitive $F_{CH4}$ to total $F_{CH4}$ at each of the survey sites (n=6 at each site). Sites U-06 and U-09 are near the swimming beach and have sandy sediments. Site U-14 is in the EC flux tower footprint.**



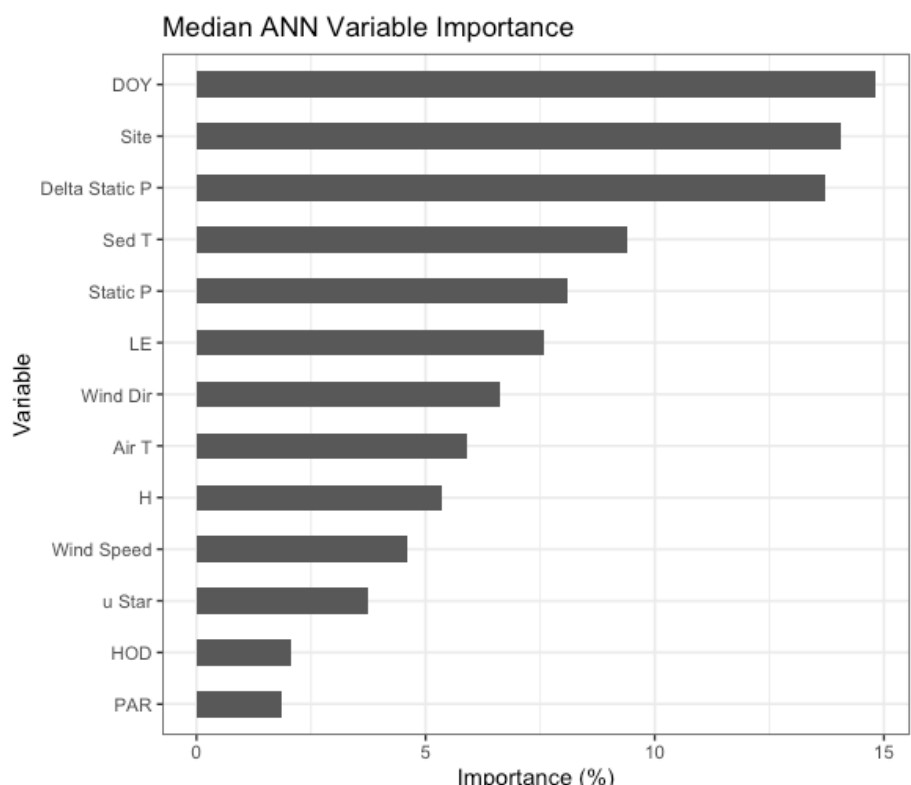

**Figure 8: Median variable importance ranking for the drivers of the artificial neural network gap-filling model in terms of percent importance to the predictive power of the model. This ranking if based both on intra-model variability (i.e. the effect of model architecture and random seed selection) and on inter-model variability (i.e. the effect of data selection for the training, testing, and validation datasets). DOY = day of year; Delta static P is change in overlying static pressure; Sed T is sediment temperature, LE is latent heat flux; Static P is static pressure; Wind Dir is wind direction; H is sensible heat flux; u Star is friction velocity; PAR is photosynthetically active radiation; HOD is hour of day.**

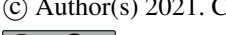


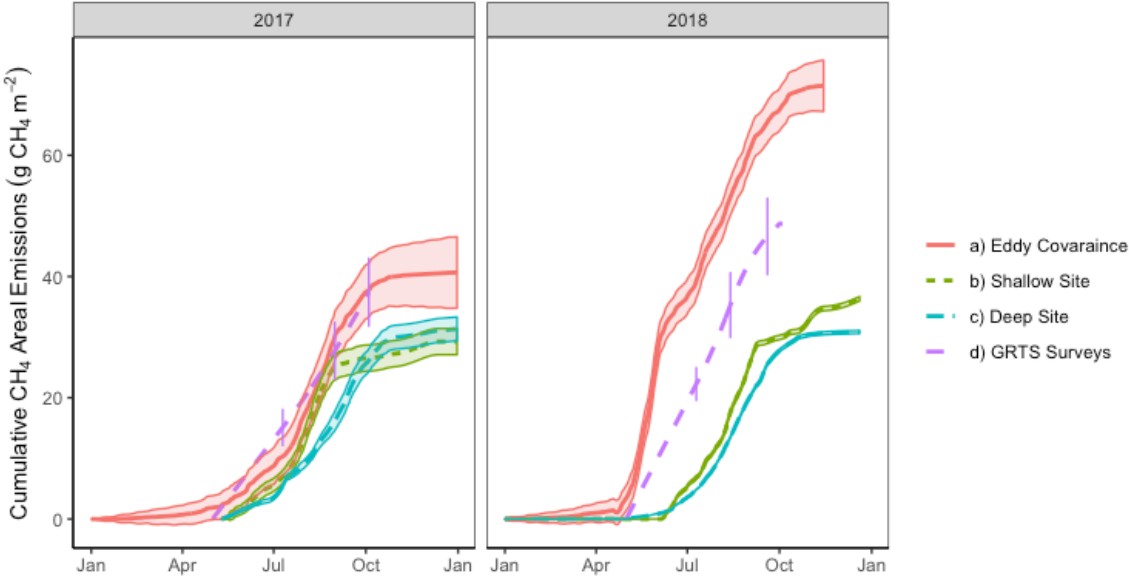

**Figure 9: Cumulative areal emissions in 2017 and 2018 from EC, sum of AFT and chamber, and spatial survey monitoring results (g CH₄ m⁻²). Vertical lines intersecting the "GRTS Surveys" trace represent the 95% confidence interval of the lake-wide F$_{CH4}$ estimate from the GRTS surveys.**



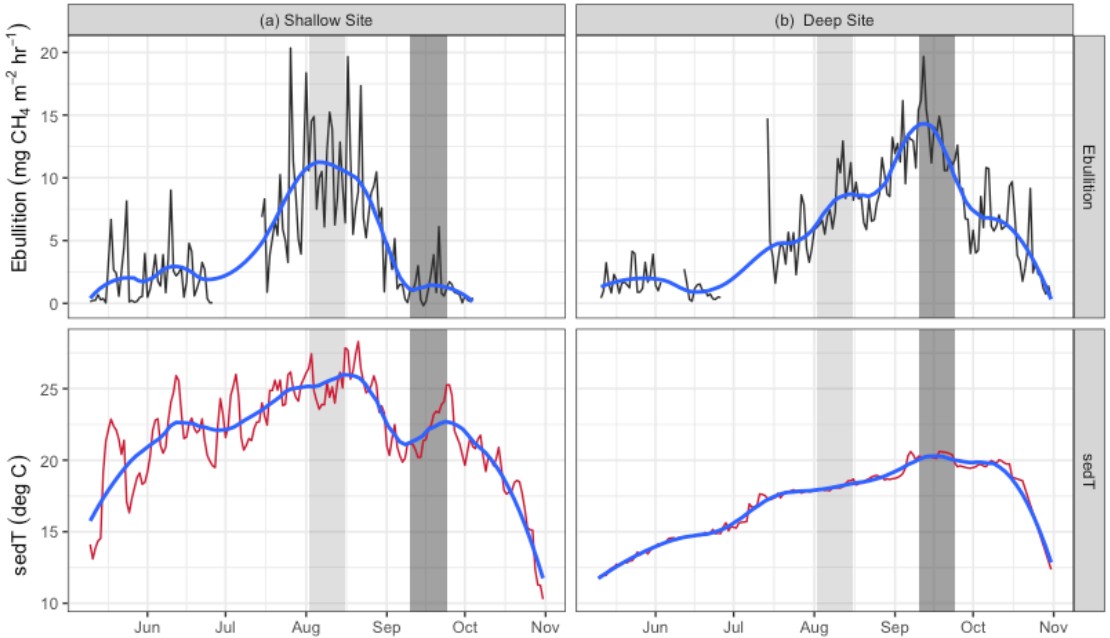

**Figure 10: Time series of sedT and ebullition in 2017 at the shallow (a, U-14) and deep (b, U-12) sites. The light grey bar highlights the period of maximum ebullition and sedT at the shallow site; the dark grey bar highlights the corresponding period at the deep site.**





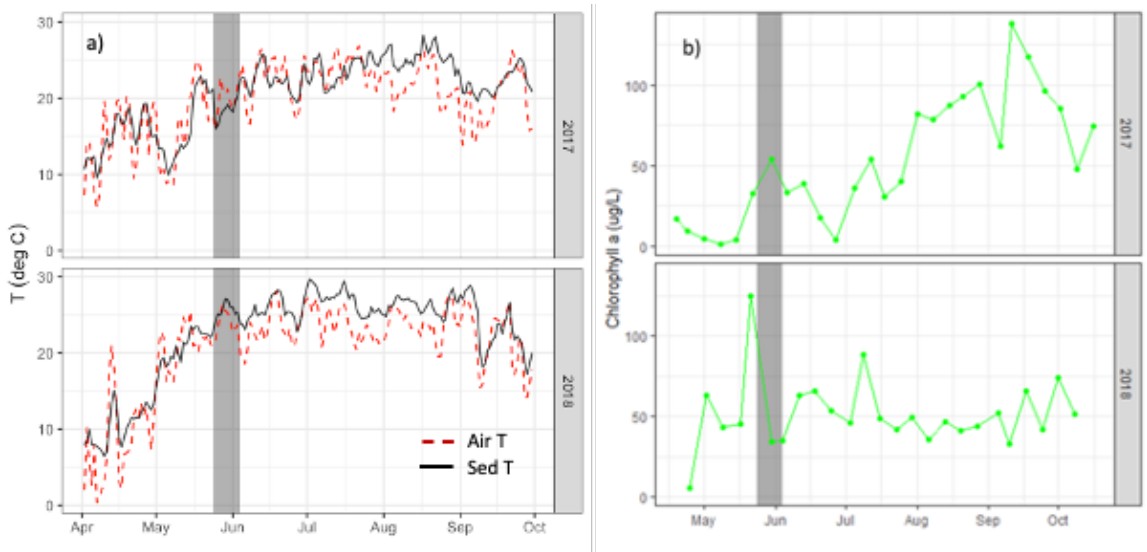

**Figure 11: Daily air and sediment temperature (a, left) and chlorophyll a (an indicator for algal biomass, b, right) in 2017 and 2018. The grey bar indicates the spring burst period of elevated FCH4 in 2018, likely supported by elevated sediment temperature and algal biomass levels that year.**


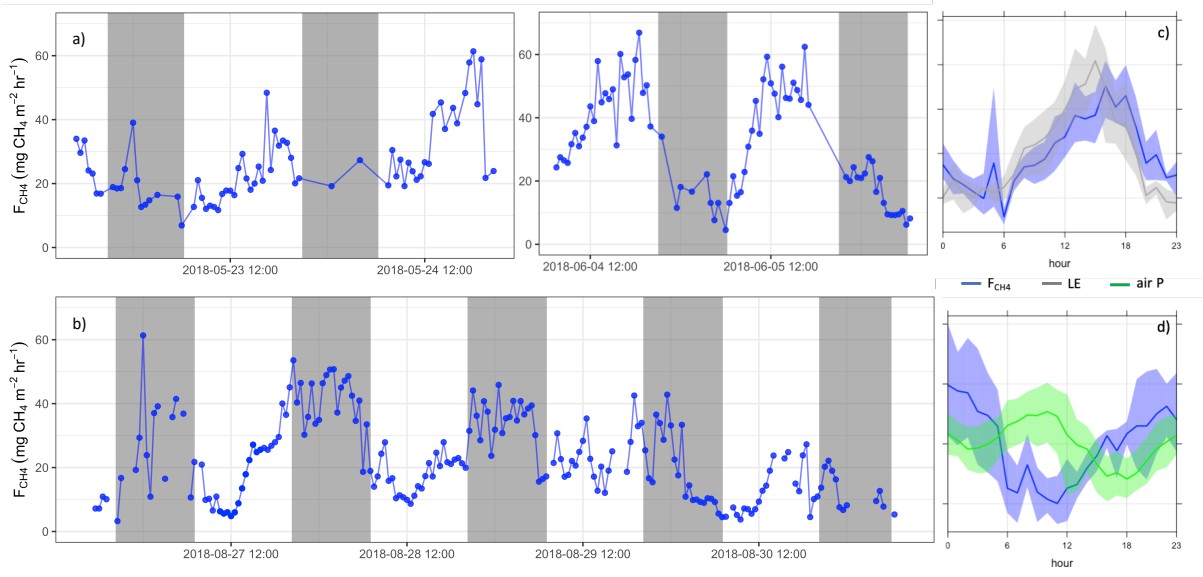

**Figure 12:** Examples of strong diurnal patterns with $F_{CH4}$ peaking during the day (a, top) and at night (b, bottom). Blue dots connected with lines in a) and b) are individual 30-minute measurements of $F_{CH4}$. Grey boxes indicate period between local sunset and sunrise. The relative mean (line) and standard deviation (shaded region) of the bin-averaged 30-minute $F_{CH4}$ values for these periods are shown in c) and d). The biophysical driver of latent heat flux (LE, grey) is also plotted in c), and air pressure (air P, green) in d).