# Peer review of "A spring burst of emissions account for most of the inter-annual and intra-lake difference in methane emissions in a small eutrophic reservoir: insights from two years of eddy covariance monitoring"

_Biogeosciences, 2021_

## Referee Comment (RC1)

**General comments**

The study provides a valuable dataset given its high spatial and temporal resolution and a variety of sampling methods, which is rare in the current literature on methane dynamics in inland waters. Therefore, the study has a good potential to offer original insights on the subject, with a unique perspective on spatial and temporal patterns and methodological biases. Data collection, curation, and interpretation are generally appropriate (although I have a limited expertise in the EC technique). However, the structure of manuscript, the presentation of the results, and the discussion around them can be vastly improved. I would like to acknowledge the hard work of the authors for producing this manuscript, and I am confident it will be suitable for publication after some modifications following external feedback.

*Goal definition*

To help the reader follow the logical structure of the study, it would be helpful to define the aims of the research in a more specific manner. The gap that the authors are filling with their research is not clearly stated in the introduction. For instance, at page 2 line 25-26: "*many questions about reservoir emission behaviour remain*" is very vague. While the authors state that they "*investigate biophysical drivers of CH$_4$*", they should be more clear about how their study differs from multiple other studies investigating CH$_4$ aquatic drivers, and how their unique dataset enables them to tackle more specific still unanswered questions on the subject. For instance, are the drivers similar at different temporal scales? How different method capture or miss those drivers and what are their biases/uncertainty when upscaling?

*Main message*

The manuscript provides a lot of scattered new information, however, the main conclusions are diluted and not clearly highlighted in the manuscript. Defining the study aim will help on that matter, but the authors also need to choose a few key results and conclusions and structure the manuscript to focus on them. The fact that the manuscript contains 3 tables and 12 figures (+10 supplemental figures) clearly reflects this issue!! Authors should select a few central figures and tables, and move the ones presenting secondary information to the supplemental document, but overall, the number of figures should be drastically reduced (main and supplementary). Accordingly, the structure of the discussion, the abstract, and the conclusion should be adapted to put the focus on the main findings.

*Results presentation and discussion*

The structure of the discussion is confusing. For instance, the first section named "*Biophysical drivers*" also outlines spatial and temporal trends, and the CH$_4$ drivers is also discussed in subsequent sections. Following previous comments, authors should find a more logical structure for discussing results. In general, the literature context for discussing the results can be improved, as the authors make little comparison with results from previous similar studies. Presentation of the results, especially in figures, should be streamlined as there is a lot of repetition.

**Specific comments**

- Line numbering should be continuous, not restarting on each new page.

- Page 2 line 17: "*in space _in_ time*" replace by and

- Page 9 line 3: "*elevated _are_ positively*" replace by and

- Page 9 line 11-12: "*The period…smaller median*" this sentence could be simplified as follows: …if 1) the difference between daytime vs nighttime $F_{CH4}$ median was >50 %.

- Section 3.1, the title of this section could be replaced by "Temporal patterns in $F_{CH4}$" since it does not only focus on seasonal trends.

- Page 9 line lines 22-26: the two sentences are repetitive and can be combined into one.

- Page 10 line 2 "*in contrast…*" and line 11-13 "*This difference…*", page 11 line 2-3 "*Much of this behaviour…*" statements like these belong in the discussion section.

- Page 10 line 20-25: Was there any investigation done concerning the $CH_4$ drivers on a day to day scale? It seems like an important component if looking at drivers at different temporal scales.

- The first paragraph of section 3.3 belongs to the method section. The second paragraph of this section could be moved to section 3.1 as it relates to the temporal measurements and drivers of $CH_4$. Also, main drivers of $CH_4$ derived from the ANN analysis should be mentioned in this result section rather than just referring to the figure.

- Section 3.4 should be restructured to present the overall budgets from different methods and comparing them before discussing the differences between years which was already discussed in section 3.1.

- The first paragraph of section 4.1 mostly contains information that belong in the method and results sections.

- Page 13 line 11-12 "*Our analysis…*" authors should be careful with this statement as they have not performed an analysis that specifically support that statement. The cited figures are only visual aids but do not include any statistical testing of this hypothesis.

- Section 4.2.1: here the authors should include a wider range of literature studies linking $CH_4$ to Chla at global spatial scales, in several temporal studies, and discussing its known link to pelagic oxic methane production.

- Section 4.2.2: When talking about diurnal $CH_4$ drivers, authors mention that non-diurnal factors may contribute to the variability in $CH_4$. While these other factors may influence $CH_4$ on different temporal scales, by definition, they do not affect its diurnal variability. Thus, I do not see the point in mentioning them when talking about diurnal variability, and the authors should hypothesize another explanation for this.

- Page 14 line 28 "*static P*" this term is not previously defined.

- Page 15 line 2-3 cumulative $F_{CH4}$ high and low here means overestimate and underestimate the mean? The sentence is not clear.

- Page 15 lines 12-14 "*In contrast…*" I don't understand this part.

- Page 15 line 15 remove "nevertheless"

- Page 15 line 18 and line 20: year missing for Deemer et al. reference.

- Page 15 lines 24-26 "*The two open-water…*" there is no context explaining what the Knox et al. (2019) analysis is. I do not understand this comparison and the conclusion drawn from it. This should be reformulated.

- Section 4.3: the last 3 paragraphs of this section are quite hard to follow and should be rewritten to improve clarity.

- Page 16 lines 14-16 "*These levels of emissions…*" in their study authors do not test any large scale pattern related to productivity or latitude so I suggest removing this statement.

- Authors do not discuss the limitations and potential biases of the EC method compared to other techniques, and do not discuss the reasons behind a more elevated flux when using this method. This should be addressed.

- Table 3, the warm-season definition should be moved to the title of the table since it applies to all presented data.

- Figure 1 legend, line 3 "*, and spatially…*" remove the "and" here.

- Figure 2, here it is difficult to compare the results of different methods, for that, I suggest putting them on top of each other in the same graph. Also, panels b and c represent only ebullition (unless the legend is wrong) while other panels are integrated $CH_4$ flux measures, so it is a bit confusing to put them in the same graph. At least, it should be clearly indicated on axis label.

- Figure 4: Panels a and b are a repetition of the same data, remove one.

- Figure 5: it is quite difficult to distinguish any spatial or temporal pattern with these types of graph. A heat map may be more suitable.

- Figure 7: this could go in the supplementary information.

- Figure 11: this could be included in figure 3, which already has the temperature time series. Also, in the legend avoid statement that belong in the discussion.

- This figure does not deliver clearly its message. Physical drivers should be plotted on top of the flux to be able to see any visual correlation between the 2, and ideally perform a statistical analysis to corroborate any correlation.

---

## Author Response (AR1)

On behalf of all authors, I would like to thank the associate editor for his feedback, and the three referees for their detailed reviews with relevant and constructive comments to improve the quality of the manuscript. Following are specific responses to the AE's comments to the author as well as to each of the referees' comments.

*The referees' comments are in black and the author's original responses are in red; updated responses pertaining to revisions made for the updated submission are in blue.*

**Associate Editor Decision: Reconsider after major revisions** (01 May 2021) by Ji-Hyung Park
Comments to the Author:
Dear Authors

I thank you for providing detailed responses to the comments and suggestions offered by three reviewers.

I agree with the three reviewers who recognized the value and novelty of your methane data obtained using a combination of various approaches. However, the reviewers also raised several critical issues including those about manuscript brevity and clarity. While your responses well addressed the reviewer comments, I envisage that the manuscript would require a substantial revision to address all raised issues and suggestions. Therefore, I have to recommend 'reconsider after major revisions' and might need to ask the reviewers to reevaluate the revised manuscript.

Thank you for acknowledging the value of our study, and for the opportunity to have the manuscript reconsidered for publication following revisions.

Regarding your response to the first reviewer's comment on the large number of figures ("Moving five figures (current Figures 4, 5, 6, 7, and 12) from the main manuscript to the supplement"), I would suggest that you consider reducing the number of the existing supplementary figures, because you would have too many figures in the supplement. You can at least combine Figs. S10 and S11 into a multi-plot figure. Please also consider preparing a cover page for the supplement to show the title and author information as well as the contents.

In this revised version, we also updated the supplementary information following this feedback. The SI now has 11 figures and a cover page.

If you want to focus the revised discussion around the spring burst, it would be very helpful to provide more supporting information about seasonal differences in nutrients and carbon components as commented by the third reviewer. In case you did not measure the relevant components except Chl a, please provide at least some secondary information from the literature.

We have added more information on seasonal and spatial differences in nutrients and carbon that show elevated levels of NH4, TP, SRP, and POC near the inflow during the spring burst (Table 3), as well as incorporating findings from the recent literature in the discussion.

I would like to ask you to submit your revised manuscript together your responses as instructed in the author guidelines.

The author's response in case of "minor" or "major" revisions must be submitted as one separate *.pdf file (indicating page and line numbers), structured in a clear and easy-to-follow sequence: (1) comments from referees/public, (2) author's response, and (3) author's changes in manuscript. Regarding author's changes, a marked-up manuscript version (track changes in Word, latexdiff in LaTeX) converted into *.pdf and combined with the author's response should be provided.

Sincerely,

Ji-Hyung Park
Associate Editor, Biogeosciences

On behalf of all authors, I would like to thank the three referees for their detailed reviews with relevant and constructive comments to improve the quality of the manuscript. Following are specific responses to each of the referees' comments.

Response to referee comment #1

General comments: The study provides a valuable dataset given its high spatial and temporal resolution and a variety of sampling methods, which is rare in the current literature on methane dynamics in inland waters. Therefore, the study has a good potential to offer original insights on the subject, with a unique perspective on spatial and temporal patterns and methodological biases. Data collection, curation, and interpretation are generally appropriate (although I have a limited expertise in the EC technique). However, the structure of manuscript, the presentation of the results, and the discussion around them can be vastly improved. I would like to acknowledge the hard work of the authors for producing this manuscript, and I am confident it will be suitable for publication after some modifications following external feedback. Thank you for acknowledging the value of our measured dataset, and for your detailed review and comments that will help us to improve the quality of the manuscript.

Goal definition: To help the reader follow the logical structure of the study, it would be helpful to define the aims of the research in a more specific manner. The gap that the authors are filling with their research is not clearly stated in the introduction. For instance, at page 2 line 25-26: "many questions about reservoir emission behaviour remain" is very vague. While the authors state that they "investigate biophysical drivers of CH4", they should be more clear about how their study differs from multiple other studies investigating CH4 aquatic drivers, and how their unique dataset enables them to tackle more specific still unanswered questions on the subject. For instance, are the drivers similar at different temporal scales? How different method capture or miss those drivers and what are their biases/uncertainty when upscaling?

We appreciate this helpful feedback which was echoed in the other reviewer's comments. We agree that the manuscript can be improved by better defining the study aims and focusing on key findings that contribute new information to the body of knowledge on aquatic CH4 emissions.

Planned changes to improve the goal definition include:
1. Changing the manuscript title to highlight the study findings: "Short-term emissions account for most of a two-fold inter-annual difference in methane emissions in a small eutrophic reservoir: insights from two years of monitoring with eddy covariance and spatial surveys" We have changed the title in the revised submission to "A spring burst of emissions account

for most of the inter-annual and intra-lake difference in methane emissions in a small eutrophic reservoir: insights from two years of eddy covariance monitoring"

2. Defining the aims of the study more clearly in the abstract and introduction. We are re-focusing the manuscript around the questions:
   1. What can we tell about the relevant importance of hot-spots vs. hot-moments to sampling bias by comparing results from different methods?
   2. How important is interannual variability in one lake (in this case, the spring burst), and what causes it?

   In the manuscript revision, we thought more about what we measured in terms of $F_{CH4}$ across the reservoir. Because we didn't see the spring burst at the deep site, we have re-focused around the following questions about combined spatiotemporal variability:
   1. How important can interannual and intra-lake variability be in a single reservoir, and what causes it?
   2. What does this tell us about how limited monitoring resources can best be used to constrain reservoir methane emissions?

Main message: The manuscript provides a lot of scattered new information, however, the main conclusions are diluted and not clearly highlighted in the manuscript. Defining the study aim will help on that matter, but the authors also need to choose a few key results and conclusions and structure the manuscript to focus on them. The fact that the manuscript contains 3 tables and 12 figures (+10 supplemental figures) clearly reflects this issue!! Authors should select a few central figures and tables, and move the ones presenting secondary information to the supplemental document, but overall, the number of figures should be drastically reduced (main and supplementary). Accordingly, the structure of the discussion, the abstract, and the conclusion should be adapted to put the focus on the main findings.

Results presentation and discussion: The structure of the discussion is confusing. For instance, the first section named "Biophysical drivers" also outlines spatial and temporal trends, and the CH4 drivers is also discussed in subsequent sections. Following previous comments, authors should find a more logical structure for discussing results. In general, the literature context for discussing the results can be improved, as the authors make little comparison with results from previous similar studies. Presentation of the results, especially in figures, should be streamlined as there is a lot of repetition.

The revised manuscript will focus on the results the three referees highlighted and the authors agree are the most important. These include the observed spring burst of FCH4, the role of sediment T, precipitation, and chla in driving the spring burst, and the difference between methods in capturing drivers and in upscaling.
In the revision, in addition to the topics listed above we also pay more attention to the intra-lake variability between shallow and deep portions of the lake, and the potential role of multiple methane production pathways.

Planned changes to the results and discussion include:

3.  Expanding the results section describing the warm-season and annual budgets to compare budgets from different methods
    We added a column of cumulative annual emissions to Table 3 for all methods including the new hybrid upscaled estimate.

4.  Clarification of how and why we use FCH4 results from the different monitoring methods to interpret different FCH4 phenomenon
    We added clarification around when we were using the EC flux results and when we were using the hybrid results

5.  Deemphasizing and reducing the discussion of FCH4 diurnal patterns and intra-reservoir spatial patterns
    We have deemphasized the discussion around diurnal patterns and distilled the discussion around intra-reservoir patterns to relate more directly to the recent literature.

6.  Moving five figures (current Figures 4, 5, 6, 7, and 12) from the main manuscript to the supplement
    The original submission had 12 Figures and 3 Tables, the new submission has 8 Figures and 4 Tables. We moved original figures 4, 6, 7, and 12 to the supplement. We kept Figure 5 and updated it to better illustrate the hybrid upscaling approach we used.

7.  Restructuring the discussion to directly address upscaling implications based on the study results. Instead of breaking the discussion into sections that still overlap (4.1: Biophysical drivers, 4.2: Temporal patterns, 4.3: Comparison with other systems and methods), we plan to structure the discussion to answer our guiding questions:
    1.  Comparison with other systems and methods
    2.  Implications for upscaling
        The revised discussion follows this structure, with the upscaling section further broken down by Spring Burst and Additional intra-lake variability

Specific Comments:

o   Line numbering should be continuous, not restarting on each new page. Thank you for pointing this out; we will use continuous line numbering in our revised manuscript. Continuous line numbering has been applied.

o   Page 2 line 17: "in space in time" replace by and OK Changed

o   Page 9 line 3: "elevated are positively" replace by and OK Changed

o   Page 9 line 11-12: "The period…smaller median" this sentence could be simplified as follows: …if 1) the difference between daytime vs nighttime FCH4 median was >50 %. OK Changed

o   Section 3.1, the title of this section could be replaced by "Temporal patterns in FCH4" since it does not only focus on seasonal trends. We agree and will make this change in the revision. This section has been changed to "3.2: Temporal patterns in $F_{CH4}$"

o   Page 9 line lines 22-26: the two sentences are repetitive and can be combined into one. OK These sentences reporting the warm-season mean airT, sedT, and LE have been removed.

o   Page 10 line 2 "in contrast…" and line 11-13 "This difference…", page 11 line 2-3 "Much of this behaviour…" statements like these belong in the discussion section. This is helpful for

guiding the process of streamlining the results and discussion in the revision.
These statements have been moved to the discussion or removed.

- o Page 10 line 20-25: Was there any investigation done concerning the CH4 drivers on a day to day scale? It seems like an important component if looking at drivers at different temporal scales. We did not directly investigate FCH4 drivers on a day-to-day scale. This scale is inherently part of the ANN gap filling model. We emphasized seasonal, interannual, and diurnal time scales because of the potential impact of biased upscaled estimates. We would expect day-to-day variability to be more stochastic.

- o The first paragraph of section 3.3 belongs to the method section. The second paragraph of this section could be moved to section 3.1 as it relates to the temporal measurements and drivers of CH4. Also, main drivers of CH4 derived from the ANN analysis should be mentioned in this result section rather than just referring to the figure. We agree and will make these changes as part of streamlining the results and discussion in the revision. We moved the first paragraph of the original section 3.3 "EC gap filling and uncertainty", to the end of methods section 2.2: Eddy covariance flux measurements. We moved the second paragraph to Results section 3.1: "Temporal patterns in $F_{CH4}$".

- o Section 3.4 should be restructured to present the overall budgets from different methods and comparing them before discussing the differences between years which was already discussed in section 3.1. We plan to rearrange the results section to present the overall budgets first, and expand this section to compare budgets from different methods We separated out section 3.2: Cumulative $F_{CH4}$. In this section, we compare budgets from different methods, including the new hybrid upscaling method.

- o The first paragraph of section 4.1 mostly contains information that belong in the method and results sections. We agree and will make this change in the revision. The information that was in this paragraph has been moved.

- o Page 13 line 11-12 "Our analysis…" authors should be careful with this statement as they have not performed an analysis that specifically support that statement. The cited figures are only visual aids but do not include any statistical testing of this hypothesis. We will rephrase this to clarify that while supported by observations, the connection from precipitation to algal biomass to FCH4 is not unequivocal. We removed the phrase "our analysis" and reworked this section to explore possible mechanistic connections and how they are supported by the observations.

- o Section 4.2.1: here the authors should include a wider range of literature studies linking CH4 to Chla at global spatial scales, in several temporal studies, and discussing its known link to pelagic oxic methane production. This section discussing drivers of the 2018 spring burst will undergo substantial revision. We will add discussion related to the following recent studies demonstrating links between chla and FCH4: Zhang et al., 2021; Bartosiewicz et al., 2021; McClure et al., 2020. Will also add discussion the potential importance of pelagic oxic methane production, citing Hartmann et al 2020. We added discussion related to these citations.

- o Section 4.2.2: When talking about diurnal CH4 drivers, authors mention that nondiurnal factors may contribute to the variability in CH4. While these other factors may influence CH4 on different temporal scales, by definition, they do not affect its diurnal variability. Thus, I do not see the point in mentioning them when talking about diurnal variability, and

the authors should hypothesize another explanation for this. We would argue that this analysis is an important contribution toward understanding the role of diurnal patterns in emissions in lentic systems, and whether FCH4 magnitudes tend to be higher or lower during the day. Thus, the lack of diurnal pattern and potential reasons behind that is just as important a result as observations of strong diurnal patterns. We plan to condense the discussion of FCH4 diurnal patterns. We plan to touch on these findings in brief as part of the implications of our findings on upscaling.

We have substantially reduced the discussion of diurnal patterns, removing the Diurnal patterns subsection and integrating the key points (e.g. diurnal patterns near the spring burst and how that relates to the emission pathway) into the other sections. Mention of other non-diurnal factors dampening the diurnal variability has been removed.

o   Authors do not discuss the limitations and potential biases of the EC method compared to other techniques, and do not discuss the reasons behind a more elevated flux when using this method. This should be addressed. We plan to address these items in more depth by expanding the comparison of methods in the results.

We have expanded the comparison of methods, most notably by combining the EC measurements with the deep AFT and spatial survey results to obtain a hybrid best estimate of lake-scale emissions.

Response to referee comment #2

General Comments: I agree with reviewer #1 on the high potential of this well conducted study on CH4 emissions from a temperate eutrophic reservoir which includes 2 years of continuous monitoring of total CH4 emissions by eddy covariance (EC) and gap-filling with ANN and ebullition with automated bubble traps at shallow and deep sites and six extensive field surveys during which diffusion (floating chambers) and ebullition (manual bubble traps) were measured at more than 10 sites. The interpretation on the spatial and temporal variability of CH4 emissions can be done on the basis of meteorology (Rainfall, temp, atmospheric pressure), energy balance (H, LE), hydrodynamics (Brunt-vaisala Freq, temp profiles), hydrology (water inputs, water levels) and biogeochemistry (O2, Chloa). Thank you for acknowledging the quality of this study, and for your detailed and constructive comments.

Major comments: My first major comment is about the result section which does not depict the whole dataset. Indeed, only CH4 fluxes are described but not correctly (see below). Information on meteorology and hydrology would be very welcomed. Description of the energy balance, thermal stratification and its spatial variability, vertical biogeochemical stratification (O2, CH4…) and their spatial variability and chlorophyll a data and its spatial variability are required.

This comment speaks to the tension between focus and thoroughness in a manuscript. We provide key information on meteorology and hydrology in results section 3.1, which are depicted in Figure 3 (temperature, LE and H, precip, stream inflow, water level, the Brunt-Vaisala frequency, and water temperature profile). Information on vertical stratification of pCH4 is provided in the supplement (Figure S4). Estimating the energy balance over open water

is challenging because of the high degree of uncertainty in the storage term. In contrast to terrestrial systems, the energy balance would have limited utility in diagnosing the quality of the EC measurements in our study. Similarly, while there are some limited data we could add about the spatial variability in dissolved nutrients and chlorophyll a, it would need to directly contribute to the main findings of the manuscript.

The revised manuscript now includes more information on the spatial variability in chla and water chemistry.

For CH4 emissions, I would recommend to separately describe ebullition (funnels, bubble traps), diffusion (floating chambers) and total emissions from EC. We plan to expand the results section describing the warm-season and annual budgets to compare budgets from different methods.

We have expanded the results to include cumulative annual emission estimates from each method. We added a column of cumulative annual emissions to Table 3 for all methods including the new hybrid upscaled estimate.

As a matter of fact, I wonder whether the gap-filling is not already a kind of interpretation as the gap-filling is based on the covariation of the fluxes with other variables when EC data are available. Therefore, it has to be decided by the authors to keep it in the result section or move it to the discussion. Independently of where the gap-filled fluxes are described (results or interpretation), it would be very informative for the reader to have information on the validated fluxes ("real data") and on the EC fluxes after gap filling for comparison. It is true that the gap-filled EC flux dataset is dependent on driver variables. For this reason, we only use the directly measured/non gap-filled EC data in the diurnal analysis, and the ecoQ10 and 2DKS analysis relating FCH4 to sediment T. We realize this is not clearly explained in the manuscript and will clarify this point in the revision. For interpreting overall patterns in FCH4, and CH4 budgets, it is better to use the gap-filled dataset, as it mitigates any bias due to data coverage.

We added this sentence to the beginning of Section 2.9: Statistical and Quantitative Analysis: For these analyses, we used the non-gap-filled measurement time series.

The second major comment is related to the absence of information regarding the calculation of total emissions from the reservoir. A critical discussion on the comparison of the different type of measurements is required in order to determine the adequate methodology to combine them for a robust estimation of total emissions. We currently ignore whether the emission factor given in the manuscript is an average of all measurements, whether it is only based on EC... Did the author take into account the bathymetry for the extrapolation of ebullition from the reservoir since ebullition at deep sites is lower than at shallow sites? We agree with this comment and plan to address this in the revision by expanding the results section that describes the budget from different methods and adding a discussion of our assumptions in estimating total reservoir emissions.

During the manuscript revision, we realized we could integrate the results from the spatially-extensive surveys with the results from the continuous measurements. We added a description of how we did this to the methods section under upscaling.

Minor comments

- Throughout the manuscript: Does "Static pressure" depict atmospheric pressure or the sum of atmospheric and hydrostatic pressure? The sum of atmospheric and hydrostatic pressure. We will specify this where we introduce static pressure as a driver of the ANN in the methods. We added this phrase to the methods: "where static pressure is the sum of overlying atmospheric and hydrostatic pressure"
- Did the author explore the role of hydrostatic pressure (water level and their variations) on CH4 emissions? Yes, as noted above, hydrostatic pressure was included as a component of static pressure.
- Did the authors attempt to decipher diffusive fluxes and ebullition from the EC dataset (at least when they have concomitant surface concentrations and or chamber measurements with EC measurements)? We used the results from the inverted funnel and chamber measurements to characterize the relative importance of these two main emission pathways (Figure 7) and found that ebullition typically accounted for > 75% of total emissions. Deciphering between the two pathways in the EC dataset based on these measurements has limited value given the high level of spatial variability. There are a few studies that use wavelet analysis to partition CH4 fluxes into diffusive and ebullitive is an emerging technique (see Iwata et al., 2018; Taoka et al., 2020), but it is outside the scope of this study to apply their novel method.
- As the manuscript require substantial rewriting/reorganization in order to properly present the dataset and better focus on key results in the discussion no detail comments are provided. Thank you for serving as a referee. We hope you will provide comments on the revised manuscript.

Response to referee comment #3:
**General Comments**

This paper deals with methane emissions in a small temperate eutrophic lake. Emissions were assessed from a variety of measurement techniques (floating chambers, submerged funnels and eddy covariance) together with some environmental parameters (sediment temperature, atmospheric pressure, heat fluxes, met data…) and a neural network (ANN) approach. The paper discusses the links between CH4 fluxes and the biophysical parameters, as well as it provides an analysis of the temporal and spatial variability of those emissions. The subject is of great interest since methane emissions from reservoirs are still poorly studied and constrained at the global scale. There are very few eddy covariance-based studies with long series (2 years) as presented here. As stated before by reviewers #1 and 2, There is no doubt that the data base gathered here is worth publication in the Biogeosciences journal. Some rearrangements would be welcome before publication. Thank you for acknowledging the value of our study, and for your helpful comments.

One of the most striking results presented here is the difference between 2017 and 2018 seasonality and cumulated emissions. Unfortunately, though well argued, there are no direct measurements of nutrients and carbon (TOC, DOC, POC, quality of OM) to support these assumptions. We must disagree with this comment. We used direct measurements of the chla concentration (e.g., Figure 11 and discussion in Section 4.2.1), which is a strong indicator for algal biomass and a widely used proxy for reservoir productivity. We do plan to revise the discussion around the spring burst away from speculating about the potential role of autochthonous C vs. allochthonous C.

We have added information on dissolved nutrient and carbon levels near the inlet and near the dam to this revision as Table 3.

Discussion on the diurnal patterns is also a bit disappointing since the results are not unequivocal. As stated above in response to RC1, our finding of dynamic diurnal patterns is an important contribution toward understanding the role of diurnal patterns in emissions in lentic systems, and whether FCH4 magnitudes tend to be higher or lower during the day. Thus, the lack of diurnal pattern and potential reasons behind that is just as important a result as observations of strong diurnal patterns. We plan to condense the discussion of FCH4 diurnal patterns. We plan to touch on these findings in brief as part of the implications of our findings on upscaling.

We have substantially reduced the discussion of diurnal patterns, removing the Diurnal patterns subsection and integrating the key points (e.g. diurnal patterns near the spring burst and how that relates to the emission pathway) into the other sections. Mention of other non-diurnal factors dampening the diurnal variability has been removed.

Authors should focus the paper on the main findings which can be supported by the data provided in the paper, and subsequently, present figures might be a little bit too numerous in that perspective of a more focused paper. This is a recurring theme in the RCs, and as stated above we plan to focus the paper in the revision and move five figures (current Figures 4, 5, 6, 7, and 12) to the supplement.

The original submission had 12 Figures and 3 Tables, the new submission has 8 Figures and 4 Tables. We moved original figures 4, 6, 7, and 12 to the supplement. We kept Figure 5 and updated it to better illustrate the hybrid upscaling approach we used.

The end of the abstract is mentioning "…there is a trade-off in intensive measurement of one water body versus short-term and/or spatially limited measurements in many water bodies", and also "The insights from multi-year, continuous, spatially extensive studies like this one can be used to inform both the study design and emission upscaling from spatially or temporally limited results". These statements are indeed interesting and I wish the paper would give clearer insights and develop more on this matter in the discussion and conclusion. We appreciate that you highlighted this section of the abstract. As stated above, we plan to directly address the difference between methods and the implications for upscaling in the revision.

The revised version of the manuscript focuses more on these aspects in the discussion and conclusion. The discussion has been rearranged into two sections: 1) comparison with other

systems and methods and 2) implications for upscaling. The conclusions state recommendations for future studies aiming to characterize CH₄ emissions from reservoirs.

Rearrangements suggested by Rev 1 and 2 would improve the paper a lot since results and discussion are all mixed together at the moment. I am particularly sensitive to the place devoted to ANN gap-filling and on the way it impacts final emission numbers.

Minor comments:

o Page 4, line13: How was used time-lapse camera in this study? The time lapse camera was used to identify periods of ice-cover. We will add this information to this section of the methods. Added.

o Page 4, line 27: there were no u* filtering at EC-S1? If so, you should argue why We did not use u* filtering at EC-S1 due to insufficient temporal coverage to determine the u* threshold. We will clarify this in the revision.
This sentence was added to the methods: We did not use $u_{star}$ filtering at EC-S1 because the temporal coverage was insufficient to determine a $u_{star}$ threshold.

o Page 5, line 33: more details are needed on the way Akaike information criterion (AIC) was used to determine fitting rate of change in the chambers. See below

o Page 6, line 10-11: vertical profile were done manually, detail procedure( how long for each level) See below

o Page 6, line 30: give more details about:" a probability design that has been shown to reduce uncertainty relative…" See below

These three comments highlight the tension in finding a balance between including adequate details in the methods and streamlining the manuscript. We can expand these sections somewhat (for example, clarifying the connection between the spatially balanced probabilistic survey and the survey sites located near the swimming beach), but we do provide the relevant references to publications with more details on these methods.

o Page 9, line 26: you should give the information that "both quantitative analyses of the relationship between FCH4 and SedT yielded statistically significant results" before implying a link between those two parameters in lines 22-24 Ok
We removed information on mean sedT from the results, but included the information on the statistical significance of the ecoQ10 and 2DKS analysis before they are discussed in section 4.2.2

o Page 11, line 3: I understand that the sandy substrate mention here was brought for recreation use (beach). Is there any point to measure fluxes at the very specific place? Yes, the probabilistic GRTS design is a hybrid between a random and gridded design. Their inclusion in the survey sites reflects our effort to characterize reservoir-wide emissions.

- Page 11, lines 23-24: comment on absolute and relative importance of each factor The variable importance factors are ranked in terms of their % importance. I'm not clear on the distinction between relative and absolute importance in this context.
- Page 11, lines 28, 29 and 30: table 3 instead of table 2 OK. The table and figure numbers have been updated.
- Page 13, line 4-5: any assessment of the mentioned transfer? The transfer in question is the transfer of heat to the deeper sediment and nutrient transfer to the deeper site, in their impact on the phase shift in FCH4 and sedT at the shallow and deep sites. The heat transfer is well documented by direct measurements. The nutrient transfer is more speculative and the reference to this will be removed. Removed.
- Page 13, line 21: any nutrients data to support the suggestion mentioned here? As mentioned above, the chla measurements are a strong indicator of algal biomass. Nutrient data has been added, and this discussion has been reworked.
- Page 13, line 26-27: any measurement of residence time and output/input of C to support this? This section of the discussion on the role of autoOC and alloOC will be reduced in the revision.
  The statement about lake metabolism has been removed.
- Page 14, line 2: is this consistent with kinetic found by Grasset et al, 2018? This section of the discussion on the role of autoOC and alloOC will be reduced in the revision.
  This speculative statement about accumulating algal biomass has been removed.
- Page 14, line 28: pattern and patterning instead of patter and pattering OK Removed.
- Page 15, line 32: detail input parameter of the model used OK. Added to the methods in the revision.
- Page 15, line 33: Del Sontro et al 2018 ref missing or is this Del Sontro et al 2016? 2018, will add the reference in the revision. Added.

References:

Bartosiewicz, Maciej , Roxane Maranger, Anna Przytulska, Isabelle Laurion, Effects of phytoplankton blooms on fluxes and emissions of greenhouse gases in a eutrophic lake, Water Research, 10.1016/j.watres.2021.116985, 196, (116985), (2021).

Grasset, C., Mendonca, R., Villamor Saucedo, G., Bastviken, D., Roland, F., & Sobek, S. (2018). Large but variable methane production in anoxic freshwater sediment upon addition of allochthonous and autochthonous organic matter: Methanogenic potential of different OC types. *Limnology and Oceanography*, *63*(4), 1488–1501. https://doi.org/10.1002/lno.10786

Hartmann, J. F., M. Gunthel, T. Klintzsch, G. Kirillin, H. P. Grossart, F. Keppler, and M. Isenbeck-Schroter. 2020. High Spatiotemporal Dynamics of Methane Production and Emission in Oxic Surface Water. Environmental Science & Technology **54**:1451-1463.

McClure, R.P., M. E. Lofton, S. Chen, K. M. Krueger, J. C. Little, C. C. Carey, The Magnitude and Drivers of Methane Ebullition and Diffusion Vary on a Longitudinal Gradient in a Small Freshwater Reservoir, Journal of Geophysical Research: Biogeosciences, 10.1029/2019JG005205, 125, 3, (2020).

Zhang, Lei, Cheng Liu, Kai He, Qiushi Shen, Jicheng Zhong, Dramatic temporal variations in methane levels in black bloom prone areas of a shallow eutrophic lake, Science of The Total Environment, 10.1016/j.scitotenv.2020.144868, 767, (144868), (2021).

---

## Referee Report (RR1)

**General comments**

The revised version of the manuscript has a much better quality compared to the previous one. The structure and clarity have particularly improved. Authors have clearly put a lot of effort in responding to the comments, and their responses are overall satisfactory. However, I would like to point out few additional comments below that should be resolved before publication:

The title could be drastically reduced. Here is a suggestion: '*Temporal trends in methane emissions from a small eutrophic reservoir: the key role of spring burst*'.

While the number of figures and tables was reduced from the previous version, it is still quite high for a standard article. Thus, some of the tables and figures can easily be moved to the supplementary section (ex: Table 1 and Figure 6). Also, in Figure 8, combining the panels for shallow and deep sites would make the message more obvious and reduce the number of figure panels.

The interpretation of the results is problematic in some cases, either due to sentence formulation or to lack of evidence from the results of the study:

- Line 447-448: the suggestion to assume zero wintertime methane flux from all temperate reservoirs based only on the results from this one studied temperate system is very unreasonable.

- Line 464: 'in contrast to the earlier assessment of age and latitude as the main drivers'. While productivity was recently identified as a methane driver, it does not mean at all that age and latitude are less important drivers.

- Line 510-511: the sentence is presented as a definite conclusion, however, alloOC and autoOC were not measured in this study, so the interpretation of this result should be presented as a potential explanation rather than a fact. Also it is not clear to which results exactly the authors refer to when saying that FCH4 was more stable (not visible in Fig 2).

- Line 514-517: 'However…(Eqn 7)' the phrasing of these two sentences suggest that there is no existing literature on the climatic drivers of methane flux, except the relation with productivity. This makes little sense and I suggest removing the sentences completely.

- Line 517-520: the results from the comparison with the predictive model contrast with the observed ones (higher observed fluxes in 2018 despite the lower mean chla), they do not align like the sentence suggests. Please reformulate to clarify the interpretation here.

- Lines 566-568: I don't understand how the authors come to the conclusion that system productivity is a more important predictor of methane compared to latitude since they studied only one system thus one latitude. Besides, latitude was previously used in the literature as a proxy for temperature for predicting methane. In their study, the authors find sediment temperature to be the best predictor of methane flux, not productivity. Thus, their interpretation of the most important driver is contradictory.

**Specific comments**

- Line 19: 'arenot' add space

- Line 62: remove 'by phytoplankton' as this is not the only hypothesized pathway for CH4 production in surface waters.

- Line 64-65: the term 'surface mixed layer CH4' is confusing, reformulate the sentence.

- Line 74: not sure what you mean by 'stochastic systems'

- Line 237: 'They surveys' should be 'The surveys'?

- Line 349: 'FCH4' the CH4 should be subscript

- Line 413-416 and 426-428: these sentences belong in the discussion section rather than the results.

- Line 443 and line 451: 'Deemer et al.' the reference lacks the date

- Line 466: representative of what? Please rephrase this sentence.

- Line 479 and 482: replace '*' by 'x'

- Line 512-514: these two sentences are very unclear

- Line 547: 'wefocus' add a space

---

## Author Response (AR2)

We appreciate the associate editor's feedback, and the additional comments from one of the three original reviewers. Following are specific responses to the AE's and referee's comments in blue; excerpts from the original and revised manuscript are *italicized blue*.

**Associate Editor Decision: Publish subject to minor revisions (review by editor)** (09 Jul 2021) by Ji-Hyung Park
Comments to the Author:
Dear Authors,

Thank you for your comprehensive revisions to incorporate the comments and suggestions offered by three reviewers.

One of the three reviewers has kindly provided a second referee report, as copied below. I agree with the reviewer on the improved structure and clarity of the manuscript.

Thank you for this feedback. The constructive comments received during the revision process has led to an improved manuscript.

However, the reviewer also noted that you could further improve the interpretation of some important results while reducing the complexity shown in the long title and the large number of tables and figures. In addition, I would like to draw your attention to editorial details, including
- Some mismatch between the clean and track-changes versions, such as "and XX% of total annual emissions" (L 33, track-changes version) We addressed this mismatch for this submission:

Line 33 in tracked-changes v1: *The main difference between years was a period of elevated emissions lasting less than two weeks in the spring of 2018, which contributed 17% of the annual emissions in the shallow region of the reservoir, and XX% of total annual emissions.*

Line 32 in tracked-changes v2: *The main difference between years was a period of elevated emissions lasting less than two weeks in the spring of 2018, which contributed 17% of the annual emissions in the shallow region of the reservoir.*

- Reference format: capitalized titles (e.g., DelSontoro et al.)

        Changed capitalized titles to sentence case.

- Inconsistent table format (Tables 1-4) and some corrections required for Table 1 (indicate the number of sites under "Spatial coverage") Table 3 (use the same unit for the three N species), and Table 4 (indicate significance levels and remove "20" if it is a typo).

        Made the table fonts consistent; changed the units in Table 3.

Your manuscript can be published after a careful revision to address the reviewer comments and editorial errors. I would like to ask you to submit your revised manuscript together your responses as instructed in the author guidelines.

The author's response in case of "minor" or "major" revisions must be submitted as one separate *.pdf file (indicating page and line numbers), structured in a clear and easy-to-follow sequence: (1) comments from referees/public, (2) author's response, and (3) author's changes in manuscript. Regarding author's changes, a marked-up manuscript version (track changes in Word, latexdiff in

LaTeX) converted into *.pdf and combined with the author's response should be provided.

Sincerely,

Ji-Hyung Park
Associate Editor, Biogeosciences

General comments The revised version of the manuscript has a much better quality compared to the previous one. The structure and clarity have particularly improved. Authors have clearly put a lot of effort in responding to the comments, and their responses are overall satisfactory. However, I would like to point out few additional comments below that should be resolved before publication:

The title could be drastically reduced. Here is a suggestion: 'Temporal trends in methane emissions from a small eutrophic reservoir: the key role of spring burst'.

Thank you for this suggestion. We adopted this title with one small change: using an indefinite article, as we only saw one spring burst: "Temporal trends in methane emissions from a small eutrophic reservoir: the key role of **a** spring burst".

While the number of figures and tables was reduced from the previous version, it is still quite high for a standard article. Thus, some of the tables and figures can easily be moved to the supplementary section (ex: Table 1 and Figure 6). Also, in Figure 8, combining the panels for shallow and deep sites would make the message more obvious and reduce the number of figure panels.

We moved Table 1 to the supplement. We combined the panels in Figure 8:

Original:

[Figure]

Revised:

[Figure]

We retained Figure 6 because it is important in illustrating the intra-lake variability, a key consideration in interpretation and upscaling of reservoir $F_{CH4}$.

The interpretation of the results is problematic in some cases, either due to sentence formulation or to

lack of evidence from the results of the study:
• Line 447-448: the suggestion to assume zero wintertime methane flux from all temperate reservoirs based only on the results from this one studied temperate system is very unreasonable.

We removed the prescriptive statement. Changed from:

> *Most studies that report $F_{CH4}$ from inland waters monitor during the warm season, with less than six months of measurements (cf. Deemer et al., 2016; DelSontro et al., 2018 Bastviken et al., 2011), and the mean $F_{CH4}$ value is then extrapolated to annual total emissions. It may however be better to assume zero $F_{CH4}$ during wintertime months for temperate reservoirs, given the very low (on the same order as the warm-season uncertainty) wintertime $F_{CH4}$ measured in this study.*

To:

> *Most studies that report $F_{CH4}$ from inland waters monitor during the warm season, with less than six months of measurements (cf. Deemer et al., 2016; DelSontro et al., 2018 Bastviken et al., 2011), and the mean $F_{CH4}$ value is then extrapolated to annual total emissions. However, we measured very low (on the same order as the warm-season uncertainty) wintertime $F_{CH4}$ in this study.*

• Line 464: 'in contrast to the earlier assessment of age and latitude as the main drivers'. While productivity was recently identified as a methane driver, it does not mean at all that age and latitude are less important drivers.

We intended this statement to convey that the Barros et al. age and latitude models predict much lower emission rates than we observed, thus something else (productivity) must be important. We changed this statement from:

> *This result strengthens the finding that midlatitude, eutrophic reservoirs in the Midwest US can support high CH4 emission rates (cf. Beaulieu at al., 2014, 2016), and also supports the emerging body of knowledge around the importance of reservoir productivity as a key indicator for FCH4 (cf. Deemer et al., 2016; West et al., 2012; DelSontro et al., 2018b) in contrast to the earlier assessment of age and latitude as the main drivers (Barros et al., 2012).*

To:

> *This result strengthens the finding that midlatitude, eutrophic reservoirs in the Midwest US can support high $CH_4$ emission rates (cf. Beaulieu at al., 2014, 2016) than would be predicted by age and latitude alone (Barros et al., 2012). The high annual $F_{CH4}$ also supports the emerging body of knowledge around the importance of reservoir productivity as a key indicator for $F_{CH4}$ (cf. Deemer et al., 2016; West et al., 2012; DelSontro et al., 2018b).*

• Line 510-511: the sentence is presented as a definite conclusion, however, alloOC and autoOC were not measured in this study, so the interpretation of this result should be presented as a potential explanation rather than a fact. Also it is not clear to which results exactly the authors refer to when saying that FCH4 was more stable (not visible in Fig 2).

We agree that this should be presented as a potential explanation, and that the statement about $F_{CH4}$ being "more stable" in 2017 is unclear. We changed this section from:

> *The difference in hydrologic regimes and subsequent availability of autoOC versus allochthonous OC (alloOC, i.e. particulate or dissolved C derived from terrestrial plant tissue) also sheds light on interannual differences beyond the spring burst. The lab study by Grasset et al. (2018) found that while additions of autoOC led to pulses of $F_{CH4}$, alloOC took longer to decompose and additions led to more gradual but sustained $F_{CH4}$. Thus, the wet spring of 2017 loaded the reservoir with slow-burning alloOC, and $F_{CH4}$ was more stable, tracking with sedT to peak emissions in early fall (Fig. 2).*

To:

> *The difference in hydrologic regimes and subsequent availability of autoOC versus allochthonous OC (alloOC, i.e. particulate or dissolved C derived from terrestrial plant tissue) may also shed light on interannual differences beyond the spring burst. The lab study by Grasset et al. (2018) found that while additions of autoOC led to pulses of $F_{CH4}$, alloOC took longer to decompose and additions led to more gradual but sustained $F_{CH4}$. Thus, the wet spring of 2017 may have loaded the reservoir with slow-burning alloOC, which could partially explain the smaller magnitude of $F_{CH4}$ pulses in 2017 compared to 2018 (Fig. 2).*

• Line 514-517: 'However…(Eqn 7)' the phrasing of these two sentences suggest that there is no existing literature on the climatic drivers of methane flux, except the relation with productivity. This makes little sense and I suggest removing the sentences completely. This was not our intent, and we have re-worked this whole paragraph to improve the messaging, see response to next comment.

• Line 517-520: the results from the comparison with the predictive model contrast with the observed ones (higher observed fluxes in 2018 despite the lower mean chla), they do not align like the sentence suggests. Please reformulate to clarify the interpretation here. We meant to convey that the predicted magnitudes compare well to the measured $F_{CH4}$, but using mean annual chla flip-flops which year had larger emissions. We have revised the whole paragraph to address this comment as well as comments on lines 512-517. The original read:

> *The implications of the spring burst phenomenon on upscaling to total FCH4 are twofold. In terms of characterizing current total reservoir FCH4, the spatial and temporal variability of the spring burst mitigate its influence. This is illustrated by comparing the lake-wide survey results to the hybrid upscaling results, which agree well in both 2017 and 2018 (Fig. 5). However, in predicting future reservoir FCH4 under changing climatic regimes, it is important to characterize underlying processes that relate 515 to the climatic drivers of precipitation and temperature. Using reservoir productivity to predict FCH4 is a good place to start: the size-productivity model (Del Sontro et al., 2018) uses annual mean chla levels to predict FCH4 (Eqn 7). Acton Lake's mean chla was higher in 2017 than 2018 (Fig. 7), and the model predicts 11.1 and 10.3 mg CH4 m-2 hr-*

*1, respectively for 2017 and 2018. These values agree well with our estimates using the hybrid upscaling approach (Table 2) but flip the finding of which year had larger CH4 emissions, which was driven by sub-annual productivity dynamics. A burgeoning body of knowledge points to 520 the importance of phytoplankton ecology on lake and reservoir CH4 production, in terms of both the amount (Hartman et al., 2020; McClure et al., 2020; Zhang et al., 2021) and type (Bartosiewicz et al., 2021). Furthermore, warmer springs have increased the frequency and intensity of cyanobacterial blooms in midwestern US reservoirs over the past two decades (Smucker et al., 2021), and continued warming will likely intensify this phenomenon. Thus, the underlying factors that led to the 2018 spring burst at Acton Lake may be more common in the future and have a greater effect on the reservoir CH4 budget.*

Revised version:

*The impact, or lack thereof, of the spring burst on reservoir-wide cumulative $F_{CH4}$ has implications for the value of higher-resolution measurements. This is analogous to the question of whether the increased complexity of process-based models improves prediction over empirical models (cf. Adams et al., 2013). While the EC monitoring results almost doubled from 2017 to 2018, the hybrid upscaled estimate had only an 11% difference (Table 2, Fig. 5). Furthermore, the cumulative $F_{CH4}$ determined via the lake-wide surveys was closer to the hybrid upscaled estimate than the EC results in 2018 (Fig. 5). Using the recent size-productivity model (Del Sontro et al., 2018) to predict $F_{CH4}$ at Acton Lake based on mean annual chla levels (Eq. 7, Fig 7) yields estimates of 11.1 and 10.3 mg $CH_4$ $m^{-2}$ $hr^{-1}$ for 2017 and 2018, respectively. These values are in the same range as the warm season mean fluxes determined via the hybrid approach for Acton Lake (Table 2). However, the model results contrast with measured results in terms of which year had higher $F_{CH4}$. Furthermore, the model results would over-estimate cumulative annual $F_{CH4}$ for Acton Lake as they do not take low wintertime emissions into account.*

*Sub-annual climatic patterns and productivity dynamics may become more important in understanding and predicting reservoir $F_{CH4}$. Recent research demonstrates how warmer springs have increased the frequency and intensity of cyanobacterial blooms in midwestern US reservoirs over the past two decades (Smucker et al., 2021), and continued warming will likely intensify this phenomenon. There is also a burgeoning body of knowledge that points to the importance of phytoplankton ecology on lake and reservoir $CH_4$ production, in terms of both the amount (Hartman et al., 2020; McClure et al., 2020; Zhang et al., 2021) and type (Bartosiewicz et al., 2021). Furthermore, Thus, the underlying factors that led to the 2018 spring burst at Acton Lake may be more common in the future and have a greater effect on the reservoir $CH_4$ budget.*

• Lines 566-568: I don't understand how the authors come to the conclusion that system productivity is a more important predictor of methane compared to latitude since they studied only one system thus one latitude. Besides, latitude was previously used in the literature as a proxy for temperature for predicting methane. In their study, the authors find sediment temperature to be the best predictor of methane flux, not productivity. Thus, their interpretation of the most important driver is contradictory.

Similar to the comment on line 468, we intended this statement to convey that the Barros et al. age and latitude models predict much lower emission rates than we observed, thus something else (productivity) must be important. We changed this statement from:

> *These levels of emissions place Acton Lake in the upper quartile of emission rates reported from reservoirs (Deemer et al., 2016), further supporting the concept that system productivity is a more important factor than latitude in predicting CH4 emission rates (Del Sontro et al., 2018).*

To:

> *These levels of emissions place Acton Lake in the upper quartile of emission rates reported from reservoirs (Deemer et al., 2016), further supporting the concept that highly productive mid-latitude reservoirs can have higher magnitude CH4 emission rates than would be predicted by age and latitude alone (Del Sontro et al., 2018).*

Specific comments

- Line 19: 'arenot' add space done
- Line 62: remove 'by phytoplankton' as this is not the only hypothesized pathway for CH4 production in surface waters. Done
- Line 64-65: the term 'surface mixed layer CH4' is confusing, reformulate the sentence. Changed to "flux from surface waters".
- Line 74: not sure what you mean by 'stochastic systems' Changed to "irregular systems"
- Line 237: 'They surveys' should be 'The surveys'? Yes, changed to 'the surveys'.
- Line 349: 'FCH4' the CH4 should be subscript Changed
- Line 413-416 and 426-428: these sentences belong in the discussion section rather than the results.
    - Line 413 – 416: *The results from the six spatial surveys indicate an inconsistent spatial pattern in FCH4 that differs from previous findings on CH4 emissions from temperate, eutrophic reservoirs which has shown that the river – reservoir transition zone near the tributary inlets tends to be a hot spot for emissions compared to the lacustrine zone (Beaulieu et al., 2014; Beaulieu et al., 2016; DelSontro et al., 2011; Tuser et al., 2017).*
        - We moved this statement to section 4.2 of the discussion: Implications for upscaling
    - Line 426 – 428: *Emission behaviour at sites U-09 and U-06 was substantially different than at other sites: these two sites had consistently low FCH4 and tend to have higher rates of CH4 diffusion than ebullition. Much of this behaviour is likely explained by the proximity of these sites to Acton Lake's swimming beach, which has a sandy substrate that likely inhibits methanogenesis at these sites.*
        - We retained this statement in the results section. This explanation of the emissions at these sites does not directly relate to any part of the study findings explored in the discussion. We think it fits better in the results as it is intended to explain why those two sites weren't investigated further.
- Line 443 and line 451: 'Deemer et al.' the reference lacks the date Added
- Line 466: representative of what? Please rephrase this sentence. Changed to: "what is representative of reality?"

- Line 479 and 482: replace '*' by 'x' done
- Line 512-514: these two sentences are very unclear
  - This relates to the reviewer comments on lines 514-517 and 517-522, above. We have reworked this paragraph to clarify our point about process-based models. The sentences in 512-514 are: *The implications of the spring burst phenomenon on upscaling to total FCH4 are twofold. In terms of characterizing current total reservoir FCH4, the spatial and temporal variability of the spring burst mitigate its influence. This is illustrated by comparing the lake-wide survey results to the hybrid upscaling results, which agree well in both 2017 and 2018 (Fig. 5).*
  Which we changed to: *The spring burst's impact, or lack thereof, on reservoir-wide cumulative FCH4 has implications for the value of higher-resolution measurements. This is analogous to the question of whether the increased complexity of process-based models improves prediction over empirical models (cf. Adams et al., 2013). While the EC monitoring results almost doubled from 2017 to 2018, the hybrid upscaled estimate had only an 11% difference (Table 2, Fig. 5). Furthermore, the cumulative FCH4 determined via the lake-wide surveys was closer to the hybrid upscaled estimate than the EC results in 2018 (Fig. 5).*
- Line 547: 'wefocus' add a space done

References:

Adams, H., Williams, A., Xu, C., Rauscher, S, Jiang, X., and McDowell, N., 2013. Empirical and process-based approaches to climate-induced forest mortality models, Frontiers in Plant Science, 4. 10.3389/fpls.2013.00438